# iDPA: Instance Decoupled Prompt Attention for Incremental Medical Object Detection

Huahui Yi [1]  Wei Xu [2]  Ziyuan Qin [3]  Xi Chen [4 5]  Xiaohu Wu [6]  Kang Li [1 7 †]  Qicheng Lao [6 †]

## Abstract

Existing prompt-based approaches have demonstrated impressive performance in continual learning, leveraging pre-trained large-scale models for classification tasks; however, the tight coupling between foreground-background information and the coupled attention between prompts and image-text tokens present significant challenges in incremental medical object detection tasks, due to the conceptual gap between medical and natural domains. To overcome these challenges, we introduce the iDPA framework, which comprises two main components: 1) Instance-level Prompt Generation (IPG), which decouples fine-grained instance-level knowledge from images and generates prompts that focus on dense predictions, and 2) Decoupled Prompt Attention (DPA), which decouples the original prompt attention, enabling a more direct and efficient transfer of prompt information while reducing memory usage and mitigating catastrophic forgetting. We collect 13 clinical, cross-modal, multi-organ, and multi-category datasets, referred to as ODinM-13, and experiments demonstrate that iDPA outperforms existing SOTA methods, with FAP improvements of 5.44%, 4.83%, 12.88%, and 4.59% in full data, 1-shot, 10-shot, and 50-shot settings, respectively. Code is available at https://github.com/HarveyYi/iDPA.git.

[1]West China Biomedical Big Data Center, West China Hospital, Sichuan University [2]School of Biomedical Engineering, Division of Life Sciences and Medicine, University of Science and Technology of China [3]Case Western Reserve University [4]Sports Medicine Center, Department of Orthopedics and Orthopedic Research Institute, West China Hospital, West China School of Medicine, Sichuan University [5]Department of Orthopedics and Orthopedic Research Institute, West China Hospital, Sichuan University [6]Beijing University of Posts and Telecommunications [7]Sichuan University Pittsburgh Institute. Correspondence to: Kang Li <likang@wchscu.cn>, Qicheng Lao <qicheng.lao@bupt.edu.cn>.

*Proceedings of the 42^nd International Conference on Machine Learning*, Vancouver, Canada. PMLR 267, 2025. Copyright 2025 by the author(s).

## 1. Introduction

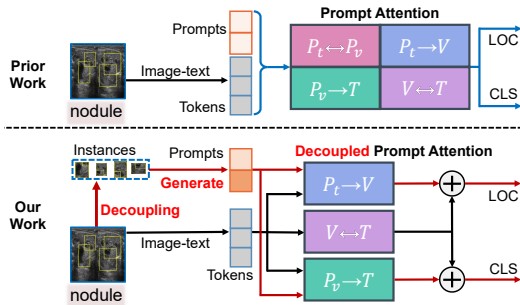

Figure 1: Comparison of our method (**iDPA**) with prior methods, highlighting improved object localization and recognition through instance-level prompt generation and decoupled prompt attention for medical detection tasks.

Vision Language Object Detection (VLOD) (Li et al., 2022; Liu et al., 2023a; Cheng et al., 2024), a paradigm that enables recognition of novel categories and scenes, has advanced object detection (OD) by integrating a language branch and leveraging large-scale image-text datasets. While these models demonstrate strong zero-shot capabilities in general domains, they struggle in the medical domain due to the degradation of medical object localization and recognition. However, neither fine-tuning separate OD models for each task nor jointly training a single model for all tasks is practical, as maintaining multiple models is inefficient and predefining all medical concepts is infeasible. Instead, Continual Learning (CL) (Li & Hoiem, 2017; Rebuffi et al., 2017; Wang et al., 2022c) is essential for adapting to emerging medical concepts while retaining prior knowledge. It must balance stability and plasticity to enable continuous learning and improve healthcare outcomes.

Recently, prompt-based CL approaches (Wang et al., 2022c;b;a; Smith et al., 2023) have gained popularity for encoding knowledge into prompt sets, enabling a frozen pre-trained model to handle sequential tasks. Compared to previous methods, these approaches not only achieve remarkable performance but also offer key advantages for continual learning. By keeping the base model unchanged and tuning only the prompt vectors, training efficiency is enhanced while eliminating the need for exemplar storage,

reducing both memory overhead and computational costs.

However, these methods are primarily designed for classification tasks and are not well-suited for OD. Unlike classification, which relies on global information, object detection demands finer-grained instance information. In previous prompt-based CL methods, as shown at the top of Fig. 1, global prompt learning incorporates both foreground and background information, which can interfere with detection tasks. Furthermore, excessive background information can confuse category recognition, especially when task modalities are similar, leading to misclassification. Additionally, prepending prompts directly to the image and text tokens dilutes the prompt information because the length of image-text tokens far exceeds those of prompts, coupling the two and hindering task-specific learning.

Given the complex attention interactions between vision and language in VLOD, the prepending approach introduces further interference between vision and text prompts. Lastly, inserting prompts into pre-trained model, *i.e.*, at the backbone level, limits the effectiveness of tuning, as fine-grained reasoning for detection occurs post-backbone, *i.e.*, at the fusion level.

In response to these challenges, we propose a framework called **i**nstance **D**ecoupled **P**rompt **A**ttention (iDPA) for incremental medical object detection. iDPA integrates Instance-level Prompt Generation (IPG) for generating fine-grained object knowledge and Decoupled Prompt Attention (DPA) to enhance prompt decoupling and precise knowledge injection during the multimodal fusion process, as shown at the bottom of Fig. 1. In the IPG module, instance features are decoupled from images, and using cross-attention to query the concept buried in the instance can separate and integrate knowledge across tasks. The DPA module decouples the originally coupled attention between prompts and tokens in previous methods, retaining three key components: vision-language mutual enhancement ($V \leftrightarrow T$), prompt-to-vision ($P_t \rightarrow V$), and prompt-to-text ($P_v \rightarrow T$) knowledge injection. Additionally, instead of injecting knowledge only at the backbone level, we also innovatively apply knowledge injection during the multimodal fusion encoder.

Consequently, iDPA incorporates three key strategies. First, by decoupling instance features from background information and incorporating fine-grained visual details into the prompt vectors, iDPA enhances object localization and recognition precision compared to randomly initialized prompts. It also effectively mitigates category confusion by focusing on the target entities and reducing spurious correlations with the background. Second, the decoupled prompt attention approach, which separates prompt vectors from token representations, accelerates knowledge injection, making it more effective for localizing and recognizing medical concepts. This also mitigates catastrophic forgetting

by preserving the original category distribution. Finally, it strategically employs the fusion encoder as the optimal knowledge injection position, which is critical for learning new medical concepts and further enhances efficiency.

In this paper, our contributions are summarized in threefold:

- We propose a novel prompt-based framework iDPA to effectively address incremental medical object detection (IMOD). It decouples instance-level knowledge and efficiently injects it into VLOD models through DPA.

- To evaluate the effectiveness of our method, we compile medical data from 13 datasets covering multiple modalities and organs, forming ODinM-13 for the IMOD task.

- Extensive experiments demonstrate the effectiveness of our proposed approach, achieving state-of-the-art performance with only 1.4% of the trained parameters in both full-data and few-shot settings.

## 2. Related Work

**Vision Language Object Detection.** Vision-Language Models (VLMs) enhance generalization by aligning visual and textual features through large-scale image-text learning. CLIP (Radford et al., 2021) and ALIGN (Li et al., 2021) leverage contrastive learning to associate images with text, inspiring GLIP (Li et al., 2022) to unify phrase grounding and object detection. Building on GLIP, MQ-Det (Xu et al., 2024) integrates a multimodal query encoder, while Grounding DINO (Liu et al., 2023a) employs a DETR-like (Carion et al., 2020) head for improved scalability. For MOD tasks, existing methods adapt pre-trained natural-domain VLOD models to the medical domain, such as MIU-VL (Qin et al.) with prompt engineering, Guo et al. (Guo et al., 2023) with prompt fusion. However, they struggle with generalization across tasks and domains.

**Continual Learning.** Continual Learning (CL) mitigates catastrophic forgetting when learning new tasks through three primary approaches. Regularization-based methods (Li & Hoiem, 2017; Kirkpatrick et al., 2017; Aljundi et al., 2018; Ding et al., 2022; Lao et al., 2021b) constrain loss functions to retain prior knowledge while adapting to new data. Architecture-based methods (Douillard et al., 2022; Li et al., 2019; Yoon et al., 2017; Mallya & Lazebnik, 2018) assign dedicated parameters for each task to prevent interference. Rehearsal-based methods (Rolnick et al., 2019; Lopez-Paz & Ranzato, 2017; Shin et al., 2017; Rebuffi et al., 2017; Lao et al., 2021a) replay stored exemplars or generate pseudo-samples to mitigate forgetting. With the rise of large-scale pre-trained models, prompt-based continual learning, an architecture-based approach, has gained prominence. L2P (Wang et al., 2022c) first introduced a prompt pool for continual learning, with DualPrompt (Wang et al., 2022b) further partitioning knowledge into general

and expert components. S-Prompt (Wang et al., 2022a) enables domain-adaptive prompt learning, while CODA-Prompt (Smith et al., 2023) improves prompt selection via attention mechanisms. DIKI (Tang et al., 2025) reduces task interference with residual tuning, and NoRGa (Le et al., 2024) models prompts as a Mixture-of-Experts with adaptive weighting. Eclipse (Kim et al., 2024) enables efficient continual panoptic segmentation via visual prompt tuning, avoiding retraining and reducing forgetting. Recently, continual learning in medical has gained increasing attention for its flexibility and adaptability to downstream tasks (Yi et al., 2023; Ye et al., 2024), offering a more practical fit for clinical use than all-in-one models like Medical SAM (Zhu et al., 2024; Ma et al., 2024).

**Continual Object Detection.** Continual Object Detection (COD) extends object detection to new categories while retaining prior knowledge. ILOD (Shmelkov et al., 2017) first introduced COD, followed by CL-DETR (Liu et al., 2023b), which improved incremental detection with distillation and memory. ZiRa (Deng et al., 2024) was the first to adapt pre-trained VLOD models for COD, mitigating forgetting through regularization and reparameterization. However, most COD research focuses on natural domains, leaving its effectiveness in data-scarce medical applications uncertain, making this an open challenge.

# 3. Preliminary

## 3.1. Task Definition

Incremental Medical Object Detection (IMOD) involves incrementally detecting and localizing medical objects (e.g., lesions, tumors, organs) in medical imaging data (e.g., CT, MRI, X-ray, PET scans) over time. The task requires sequential learning from multiple tasks $[\mathcal{T}_1, \mathcal{T}_2, \cdots, \mathcal{T}_N]$, where each task $\mathcal{T}_i$ consists of a dataset $\mathcal{D}_i = \{x_i^j, y_i^j\}_{j=1}^{N_i}$, with $x_i^j$ representing images and $y_i^j$ including bounding boxes and class labels. Each task also includes a class name set $C_i = \{c_i^j\}_{j=1}^{N_{C_i}}$, linking label indices to category names used by the text encoder of VLOD models. The main challenge in IMOD, particularly in class-incremental learning, is to adapt to new object classes introduced in each task without forgetting previously learned ones, allowing the model to handle an expanding range of medical objects while maintaining detection accuracy across all learned classes. This work is developed based on pre-trained VLOD models (such as GLIP (Li et al., 2022)). When training task $t$, the task's classes encompass the current task's classes along with the previous tasks' classes.

## 3.2. Vision Language Object Detection

To better achieve IMOD, this work builds upon pre-trained VLOD models in natural domains, providing a strong foundation for improving generalization and robustness in data-scarce scenarios, making them highly suitable for practical medical settings. Unlike traditional object detectors, VLOD models replace the classification head with a textual encoder, such as BERT (Devlin, 2018), and introduce a cross-modality fusion encoder that enhances the model's ability to detect medical objects across different imaging modalities. VLOD models for object detection consist of four key components: 1) Visual Encoder $\Phi_v$, 2) Textual Encoder $\Phi_t$, 3) Cross-Modality Fusion Encoder $\Phi_f$, and 4) Localization Head $\Phi_{\mathrm{loc}}$.

$$f_v = \Phi_{\mathbf{v}}(\text{Img}), \quad f_t = \Phi_{\mathbf{t}}(\text{Text}), \tag{1}$$

$$f_v', \ f_t' = \Phi_{\mathbf{f}}(f_v, \ f_t), \tag{2}$$

$$p_{\mathrm{loc}} = \Phi_{\mathrm{loc}}(f_v'), \qquad p_{\mathrm{cls}} = f_v' \cdot (f_t')^{\mathrm{T}}. \tag{3}$$

The workflow is as follows: First, visual and textual features are extracted using $\Phi_v$ and $\Phi_t$ (Eq. (1)). These features are then fused through the cross-modality fusion encoder $\Phi_f$ (Eq. (2)). Finally, localization $p_{\mathrm{loc}}$ and classification $p_{\mathrm{cls}}$ predictions are generated (Eq. (3)), where · denotes matrix multiplication. The fusion of these features enhances object localization and recognition accuracy.

## 3.3. Prompt-Based Continual Learning

Prompt-based CL, a type of CL method based on prompting pre-trained models, incrementally learns and stores a lightweight, learnable parameter (known as a prompt) for each task, gradually building a "prompt pool" $P = \{p_1, p_2, \cdots, p_N\}$, where $p_i \in \mathbb{R}^{l \times d}$. Here, $N$ represents the number of tasks, $l$ is the prompt length, and $d$ is the feature embedding dimension. At inference time, a selected prompt from the prompt pool is appended to the frozen pre-trained model to restore learned knowledge. Given the feature embeddings $f_e \in \mathbb{R}^{L \times d}$ for a transformer layer, the input is formed by concatenating the selected prompt $p_s \in \mathbb{R}^{l \times d}$ with $f_e$ as follows:

$$\text{Transformer}([p_s; f_e]), \quad \text{where} [p_s; f_e] \in \mathbb{R}^{(l+L) \times d}, \tag{4}$$

where $p_s$ is the selected prompt embedding. The prompt selection process relies on query-key matching, with feature centroids $\mathbb{K} = \{K_i\}_{i=1}^N$ learned during training via cosine similarity or clustering. For a test sample $x$, the most relevant centroid $K_s$ is identified by:

$$K_s = \arg \max_{K_i \sim \mathbb{K}} \langle \Phi_{\mathbf{v}}(x), K_i \rangle. \tag{5}$$

Currently, prompt-based CL methods focus on global, mixed knowledge, which suffices for classification tasks. However, for IMOD, fine-grained knowledge is crucial for precise localization and understanding of medical objects. Therefore, adapting prompt-based CL for IMOD requires focusing on learning and preserving specific, fine-grained knowledge to address the unique challenges of IMOD.

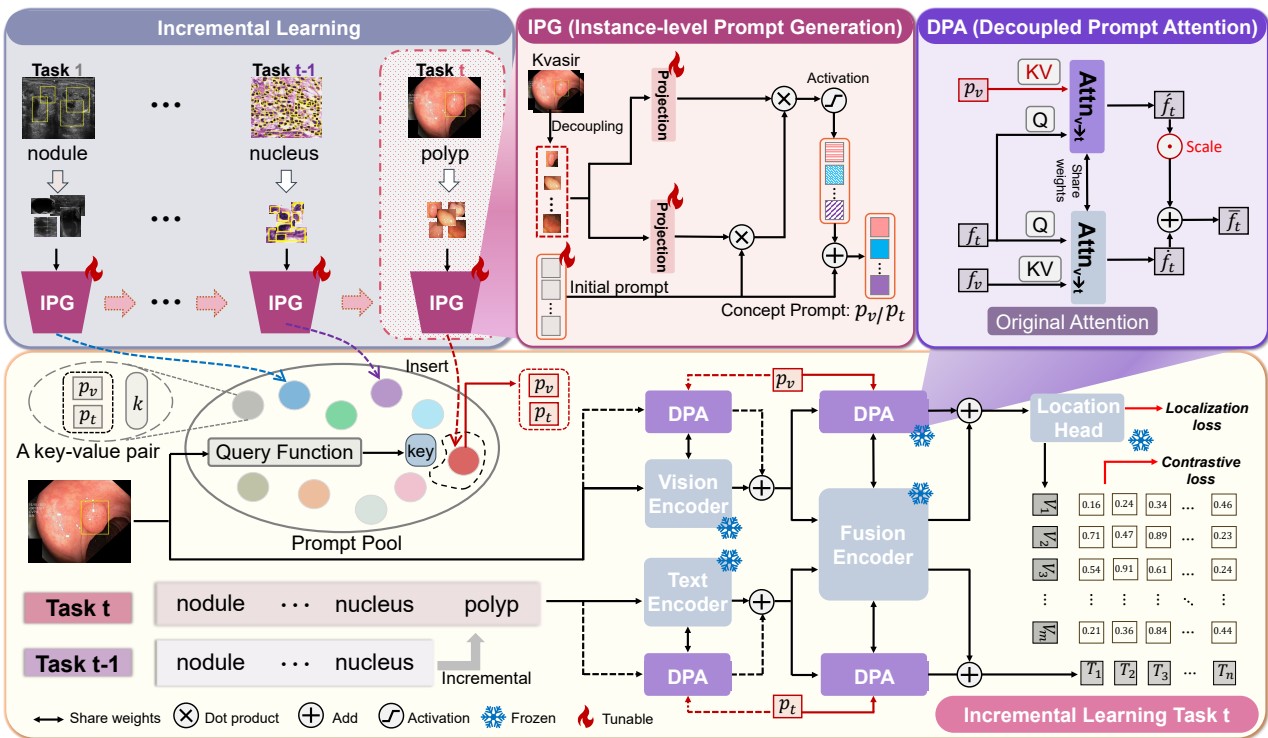

Figure 2: **Overview of iDPA**. Based on a frozen pre-trained VLOD model with visual-language interaction modules (e.g., GLIP (Li et al., 2022)), iDPA integrates Instance-level Prompt Generation (IPG) and Decoupled Prompt Attention (DPA) to enhance object localization and recognition, optimizing knowledge transfer for medical detection tasks.

## 4. Methodology

### 4.1. Overview of iDPA

To effectively achieve the IMOD goal, our core idea is to *decouple fine-grained instance-level knowledge, generate enriched concept prompts, and optimize the Prompt Attention (PA) mechanism by retaining key attention components, enabling efficient knowledge injection into the pre-trained model while mitigating class forgetting issues.* This reliable, robust approach helps the model focus on essential localization and recognition for clinical medical object detection. Thus, we propose iDPA, an efficient, scalable IMOD framework, as shown in Fig. 2.

To realize this design, iDPA integrates Instance-level Prompt Generation (IPG, Sec. 4.2) and Decoupled Prompt Attention (DPA, Sec. 4.3) to enhance robust, scalable incremental learning. First, IPG extracts fine-grained, adaptive instance-level features from the training set and generates rich, diverse, stable instance-specific prompt knowledge. This contextual knowledge is then injected into the frozen, pre-trained model through DPA, enabling the model to retain focus on reliable, critical, task-specific details while effectively mitigating interference from previous tasks. Through this streamlined process, iDPA facilitates efficient, precise and seamless fine-grained knowledge transfer, which is essential for accurate IMOD performance.

### 4.2. Instance-Level Prompt Generation

Inspired by prior research (Wu et al., 2023; Xu et al., 2024), we first use a pre-trained model to extract image features from the training set, then focus on target regions via bounding boxes. To refine these features, we apply cross-attention to disentangle and clarify different concepts, as demonstrated in (Alayrac et al., 2022; Xu et al., 2024).

**Decoupling Instance Features for Prompt Construction.**
For each task $\mathcal{T}_i$, prior to model training, we decouple the instance-level representations $\mathcal{I}_i$ for each of the $|C_i|$ categories from the training data as follows:

$$v_c^{(j)} = \text{RoIPool}(\Phi(\text{Img}, \text{Text}), \gamma b), \quad j = 1, 2, \ldots, M, \quad (6)$$

$$\mathcal{I}_i = \{\mathbf{v}_c \mid \mathbf{v}_c = \{v_c^{(j)}\}_{j=1}^M\}_{c=1}^{|C_i|}, \quad (7)$$

where each $\mathbf{v}_c \in \mathbb{R}^M$ consists of $M$ instance-level representations of each category, which are encoded by the image encoder $\Phi_\mathbf{v}$ or the fusion encoder $\Phi_\mathbf{f}$. These representations are extracted before the attention layers, which correspond to the attention layers where prompt learning is applied. Specifically, given an instance from the $c$-th category with bbox $b \in \mathbb{R}^4$ in an image, an RoI pooler (Ren, 2015) is used to extract the corresponding region feature $v_c^{(j)} \in \mathbb{R}^d$. The scaling factor $\gamma = 1.3^2$ increases the region size to capture additional contextual information. During training, for each class, we query the $M$ instance representations to decouple the $l$ concepts (note that $l$ is the prompt length

defined in Eq. (4)) contained within them, which are then used to form the prompt for the current task. In practice, we set $M = 1000$ for full-data settings to ensure diverse concept coverage. For few-shot learning, we set $M = m$, where $m$ is the number of available shots per class.

**Continual Concept Perception and Knowledge Integration (CCPKI).** Given the instance-level representations $\mathcal{I}_i = \{\mathbf{v}_c \mid |\mathbf{v}_c| = K\}_{c=1}^{|C_i|}$, extracted from the training data of the current task for each category, the CCPKI module decouples the concepts from these instances using a *Query-Answer* framework. Specifically, the generation of the $i$-th prompt $\ddot{p}_i$ can be expressed as follows:

$$\dot{p}_i = \text{softmax}\left(\frac{p_i(\mathcal{W}_k \mathcal{I}_i)}{\sqrt{d}}\right)(\mathcal{W}_v \mathcal{I}_i); \; \ddot{p}_i = p_i + \alpha \cdot \sigma(\tau \cdot \dot{p}_i). \tag{8}$$

To generate task-specific prompts from instance features, we adopt the following attention-based formulation. Here, $p_i \in \mathbb{R}^{l \times d}$ represents the initial prompt for the $i$-th task, consisting of $l$ learnable concept components that serve as queries. The instance-level representations $\mathcal{I}_i$ are projected into key and value vectors using learnable matrices $\mathcal{W}_k$ and $\mathcal{W}_v$, respectively. A cross-attention mechanism is then applied to extract task-relevant conceptual knowledge from the instances. Since different tasks may involve different concepts, a learnable scaling factor $\tau \in \mathbb{R}^{l \times 1}$ is used to dynamically modulate the concept weights, followed by a nonlinear activation function $\sigma(\cdot)$ (e.g., $\tanh$) to filter and enhance the meaningful components. Finally, the activated concepts are scaled by $\alpha \in \mathbb{R}^{1 \times d}$ and added to the initial prompt via a residual connection.

To transfer knowledge effectively, it is common for previous tasks to assist in learning subsequent ones. Inspired by this, the CCPKI for the $i$-th task is initialized with the parameters from the previous CCPKI, i.e., $\Phi_i^{CCPKI} \leftarrow \Phi_{i-1}^{CCPKI}$. In this way, our approach retains the generated concept knowledge $\hat{p}_i$ from the decoupled instances in the prompt pool after training, resulting in a dynamically evolving prompt pool. This design enhances scalability and flexibility, making it particularly well-suited for IMOD tasks.

### 4.3. Decoupled Prompt Attention

Instead of training a series of prepended prompt vectors for each task, we focus on modifying the attention mechanism to learn multimodal knowledge efficiently. Specifically, we decouple the PA mechanism into two components: the original attention and the attention with prompt knowledge injection. These components are then integrated through a residual connection. This decoupling process is referred to as the DPA mechanism. Compared to PA, DPA accelerates instance-level knowledge learning, reduces task interference, and lowers computational complexity, resulting in reduced memory usage during training.

The following outlines the derivation from PA to DPA, using the visual-language interaction module $X$-Attn (Li et al., 2022; Liu et al., 2025) as an example. The input multimodal tokens $f_v \in \mathbb{R}^{L_v \times d}$ (for vision) and $f_t \in \mathbb{R}^{L_t \times d}$ (for text) are fed into $X$-Attn, where the mutual enhancement of multimodal knowledge produces updated $\widetilde{f}_v$ and $\widetilde{f}_t$:

$$\begin{aligned}
\{\widetilde{f}_t, \widetilde{f}_v\} &= X\text{-Attn}(f_t, f_v) \\
&= \{f_t + \text{Attn}_{v \to t}(f_v, f_t), \; f_v + \text{Attn}_{t \to v}(f_t, f_v)\} \\
&= \{f_t + \bar{f}_t, f_v + \bar{f}_v\}.
\end{aligned} \tag{9}$$

In Eq. (9), $\text{Attn}_{v \to t}(f_v, f_t)$ represents the vision-to-text knowledge transfer, denoted as $\bar{f}_t$, and vice versa for the text-to-vision transfer $\bar{f}_v$. Inspired by (He et al., 2021),

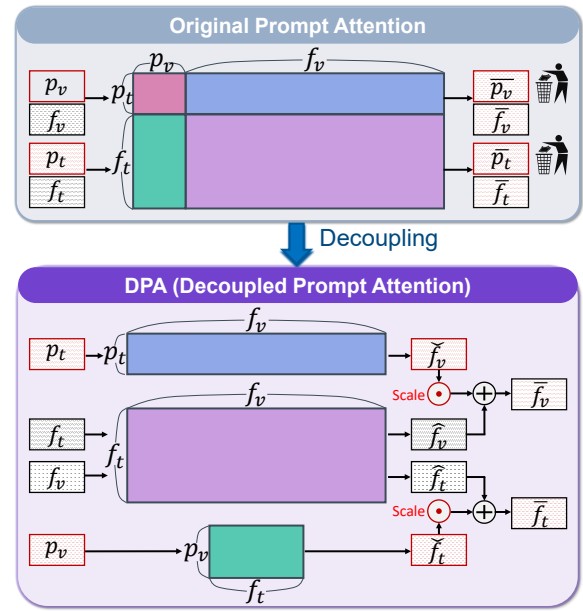

Figure 3: Comparison between Decoupled Prompt Attention (DPA) and Original Prompt Attention.

we analyze the role of prompt tuning in multimodal fusion through formal derivation. We derive an equivalent formulation to closely examine the prompt's role in $\text{Attn}_{v \to t}$ (For simplicity, we focus on $\text{Attn}_{v \to t}$, which can similarly be extended to $\text{Attn}_{t \to v}$), providing an alternative perspective of multimodal prompt tuning:

$$\begin{aligned}
&[\bar{p}_t; \bar{f}_t] = \text{Attn}_{v \to t}([p_v; f_v], [p_t; f_t]) \\
&= [(1 - \lambda(p_t))\text{Attn}_{v \to t}(f_v, p_t) + \lambda(p_t)\text{Attn}_{v \to t}(p_v, p_t); \\
&\quad (1 - \lambda(f_t))\text{Attn}_{v \to t}(f_v, f_t) + \lambda(f_t)\text{Attn}_{v \to t}(p_v, f_t)],
\end{aligned} \tag{10}$$

where $p_{\{v,t\}} \in \mathbb{R}^{l \times d}$ represent vision and text prompts, respectively. These prompts are prepended to the visual and textual features $f_v$ and $f_t$ before being fed into $\text{Attn}_{v \to t}$.

By examining Eq. (10), we introduce the DPA mechanism, which decouples the knowledge transfer processes $p_v \to f_t$ and $f_v \to f_t$ in PA. This is achieved through: 1) discarding the learning of $[\bar{p}_t]$ similar to VPT-Deep (Jia et al., 2022), which can be replaced by new trainable prompts in subsequent $X$-Attn layers to reduce computational complexity

Table 1: Performance of various continual learning methods on the ODinM-13 benchmark under the full data setting. **FAP** (%): Final Average AP. **CAP** (%): Cumulative Average AP. **FFP** (%): Final Forgetting Percentage of old tasks.

| Methods | DFUC | Kvasir | OpticN | BCCD | CPM-17 | BreastC | TBX11K | KidneyT | Luna16 | ADNI | Meneng | BreastT | TN3k | FAP ↑ | CAP ↑ | FFP ↓ |
|---|---|---|---|---|---|---|---|---|---|---|---|---|---|---|---|---|
| Zero-shot | 4.60 | 12.30 | 3.20 | 3.70 | 0.10 | 1.00 | 0.00 | 1.30 | 0.00 | 0.00 | 4.40 | 4.50 | 5.50 | 3.12 | - | - |
| SAM2.1 | 0.06 | 1.71 | 0.00 | 8.83 | 18.00 | 0.15 | 0.00 | 0.67 | 0.23 | 0.00 | 12.54 | 0.08 | 0.22 | 3.27 | - | - |
| MedSAM-2* | 4.15 | 5.34 | 4.53 | 13.83 | 22.44 | 1.54 | 0.00 | 1.56 | 4.32 | 0.04 | 18.56 | 2.96 | 3.45 | 5.02 | - | - |
| FT (Oracle) | 51.74 | 76.70 | 79.78 | 62.63 | 37.45 | 54.08 | 36.76 | 66.73 | 38.52 | 49.22 | 75.35 | 45.39 | 61.72 | 56.62 | - | - |
| Joint (Upper) | 46.82 | 72.19 | 78.70 | 61.74 | 37.18 | 53.34 | 35.35 | 65.72 | 34.03 | 47.97 | 73.90 | 43.89 | 59.90 | 54.67 | - | - |
| **Non-Prompt-based CL** | | | | | | | | | | | | | | | | |
| Sequential | 0.00 | 0.00 | 0.00 | 0.00 | 3.18 | 0.00 | 0.00 | 0.00 | 0.00 | 32.84 | 0.00 | 0.00 | 21.18 | 4.40 | 15.87 | 57.81 |
| WiSE-FT | 10.47 | 45.93 | 1.21 | 16.23 | 7.52 | 6.59 | 0.20 | 10.25 | 0.13 | 0.03 | 13.11 | 14.65 | 13.06 | 10.72 | 26.18 | 18.60 |
| ER | 34.86 | 60.04 | 54.07 | 55.72 | 15.49 | 40.90 | 17.33 | 48.23 | 13.45 | 43.93 | 65.49 | 22.15 | 47.14 | 39.91 | 48.73 | 19.25 |
| ZiRa | 0.14 | 0.13 | 0.00 | 0.71 | 0.10 | 0.42 | 0.00 | 0.00 | 0.00 | 26.85 | 0.00 | 0.15 | 19.02 | 3.66 | 16.37 | 49.67 |
| **Prompt-based CL** | | | | | | | | | | | | | | | | |
| L2P | 42.11 | 70.16 | 46.58 | 57.67 | 1.62 | 48.09 | 26.96 | 54.60 | 27.36 | 38.05 | 25.58 | 29.44 | 50.25 | 39.88 | 46.04 | 8.24 |
| DualPrompt | 44.26 | 40.47 | 17.73 | 34.03 | 0.59 | 37.03 | 5.91 | 31.29 | 23.12 | 42.30 | 15.58 | 33.57 | 49.63 | 28.89 | 42.24 | 20.57 |
| S-Prompt | 43.48 | 63.02 | 47.49 | 35.52 | 8.37 | 39.51 | 27.96 | 57.79 | 25.82 | 40.67 | 65.02 | 29.61 | 48.97 | 41.02 | 46.70 | 8.87 |
| CODA | 42.01 | 70.63 | 58.78 | 41.96 | 4.51 | 47.62 | 26.50 | 58.36 | 22.48 | 32.99 | **65.71** | 27.04 | 48.39 | 42.08 | 49.78 | 2.80 |
| DKTI | 46.91 | **75.91** | 54.14 | 55.12 | 0.74 | 47.57 | 32.63 | 62.03 | 29.21 | 43.29 | 16.02 | 34.35 | **54.77** | 42.51 | 49.12 | 7.74 |
| NoRGa | 44.62 | 75.87 | 58.44 | 57.73 | 1.07 | 51.76 | 29.71 | 63.48 | 29.53 | **44.54** | 37.56 | 34.11 | 54.46 | 44.84 | 49.90 | 4.92 |
| iDPA(Ours) | **47.09** | 73.76 | **66.85** | 60.29 | **36.54** | 50.98 | **32.69** | 64.98 | 31.15 | 44.42 | 57.20 | **34.65** | 53.03 | **50.28** | **54.10** | **2.48** |

**Notes: SAM2.1** denotes auto-segmentation using the SAM2.1-L model, while **MedSAM-2*** incorporates MedSAM-2's memory bank based on SAM2.1.

and memory usage; and 2) re-normalizing the attention weights through the weight adjustment scalar $\lambda(f_t)$, which aims to remove the coupling between the visual features $f_v$ and the visual prompt $p_v$:

$$\lambda(f_t) = \frac{\sum_i \exp(\frac{f_t \mathcal{W}_q (p_v \mathcal{W}_k)^{\mathrm{T}}}{d})_i}{\sum_i \exp(\frac{f_t \mathcal{W}_q (p_v \mathcal{W}_k)^{\mathrm{T}}}{d})_i + \sum_j \exp(\frac{f_t \mathcal{W}_q (f_v \mathcal{W}_k)^{\mathrm{T}}}{d})_j}, \tag{11}$$

where $\mathcal{W}_{\{q,v\}} \in \mathbb{R}^{d \times d}$ are projection layers. Our proposed DPA mechanism is shown in Fig. 3, and the process of knowledge transfer and fusion for the enhanced text features $\bar{f}_t$ (note that $\bar{p}_t$ is omitted) is then updated to as follows:

$$\bar{f}_t = \mathrm{Attn}_{v \to t}(f_v, f_t) + \frac{\lambda(f_t)}{1 - \lambda(f_t)} \mathrm{Attn}_{v \to t}(p_v, f_t), \tag{12}$$

$$= \mathrm{Attn}_{v \to t}(f_v, f_t) + \lambda \mathrm{Attn}_{v \to t}(p_v, f_t), \tag{13}$$

where $\lambda \in \mathbb{R}^{1 \times d}$ is a learnable parameter, initialized to 0, ensuring that the additional information in $p_v \to f_t$ does not affect the original branch $f_v \to f_t$ before training on downstream datasets. For $\lambda(f_t)$, we argue that in object detection tasks, $f_v$ is typically much larger than $p_v$ (e.g., $f_v$ may exceed 10,000 tokens, while $p_v$ is typically set to 10), causing the prompt information overshadowed by the input features, reducing learning efficiency. This imbalance in the original PA hinders the model's effective use of the prompts, impeding knowledge transfer and task performance. Compared to PA, DPA reduces the interference between prompt information and the pretrained model by lowering their coupling. Specifically, the normalized attention weight $1 - \lambda(f_t)$ before $\mathrm{Attn}_{v \to t}(f_v, f_t)$ is removed, preserving the full pre-trained model information.

The weight $\lambda(f_t)$ before $\mathrm{Attn}_{v \to t}(p_v, f_t)$ is replaced with a learnable scaling factor, providing greater flexibility in learning prompt knowledge and enabling the model to capture more precise and richer downstream information. Additionally, the decoupling process also reduces the computational complexity and memory usage of attention, as shown in 5.4.

## 5. Experiments

### 5.1. Experimental Setup

**Benchmark.** To ensure a comprehensive evaluation, we collected 13 MOD tasks (Jha et al., 2020; Boccardi et al., 2015; Cassidy et al., 2021; Liu et al., 2020; Gong et al., 2021; Setio et al., 2017; Vu et al., 2019) from publicly available datasets for IMOD, named **ODinM-13**. This benchmark evaluates model performance in real medical scenarios, covering 8 imaging modalities across 9 organs. Each task is assessed in both full and few-shot settings ($k = 1, 10, 50$), ensuring each class has at least $k$ objects. To further evaluate generalizability, we supplement ODinM-13 with a domain-incremental benchmark constructed from four polyp datasets (Ji et al., 2022; Jha et al., 2020; Bernal et al., 2015; Ngoc Lan et al., 2021) across different medical centers. For more details, please see the appendix.

**Evaluation Metric.** To evaluate the model's continual learning and forgetting mitigation, we use Final Average AP (**FAP**) for final performance, Cumulative Average AP (**CAP**) for overall performance, and Final Forgetting Percentage (**FFP**) to measure resistance to forgetting old tasks. The prediction performance for task $\hat{i}$ after learning task $i$ is denoted as $\mathrm{AP}_{i,\hat{i}}$, where Average Precision (AP) is the standard

Table 2: Overall performance of various continual learning methods on the ODinM-13 benchmark under few-shot data settings.

| Method | 1-shot | | | 10-shot | | | 50-shot | | |
|---|---|---|---|---|---|---|---|---|---|
| | FAP ↑ | CAP ↑ | FFP ↓ | FAP ↑ | CAP ↑ | FFP ↓ | FAP ↑ | CAP ↑ | FFP ↓ |
| Joint (Upper) | 20.03 | - | - | 36.68 | - | - | 46.07 | - | - |
| Sequential | 1.24 | 11.49 | 23.43 | 1.66 | 13.67 | 36.51 | 3.38 | 14.95 | 46.86 |
| WiSE-FT | 6.16 | 14.62 | 9.25 | 9.47 | 21.03 | 12.98 | 10.24 | 23.34 | 16.03 |
| ZiRa | 6.98 | 13.59 | 11.68 | 10.90 | 16.19 | 15.12 | 3.49 | 15.78 | 37.28 |
| L2P | 3.25 | 7.18 | **3.14** | 5.16 | 9.63 | 4.48 | 27.91 | 35.20 | 5.66 |
| DualPrompt | 7.36 | 13.30 | 7.93 | 5.95 | 13.99 | 11.34 | 20.44 | 34.94 | 16.91 |
| S-Prompt | 3.34 | 8.90 | 5.66 | 8.39 | 12.98 | **4.36** | 22.76 | 30.86 | 7.92 |
| CODA | 6.89 | 13.96 | 4.57 | 4.41 | 14.20 | 12.60 | 31.75 | 40.86 | 4.37 |
| DIKI | 5.67 | 13.09 | 3.88 | 6.46 | 15.34 | 9.78 | 34.06 | 42.27 | 5.85 |
| NoRGa | 6.02 | 11.09 | 5.17 | 5.36 | 11.90 | 8.14 | 32.09 | 39.12 | 5.70 |
| iDPA (Ours) | **12.19** | **18.03** | 3.58 | **23.78** | **29.68** | 4.75 | **38.65** | **45.03** | 3.93 |

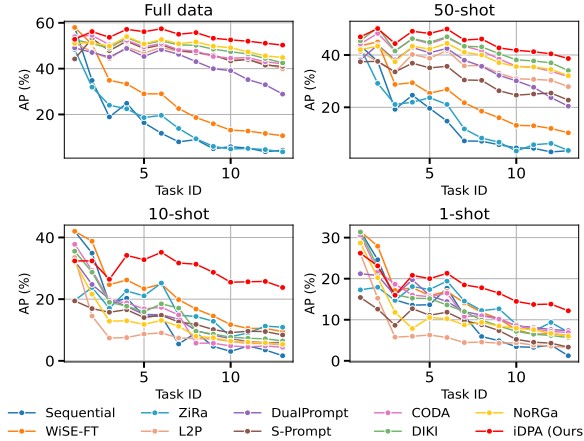

Figure 4: Performance variation of different CL methods in full-data and few-shot settings.

Table 3: Comparison on four polyp datasets under the continual domain setting.

| Methods | Sun | Kvasir | BKAI | ClinicDB | FAP ↑ | CAP ↑ | FFP ↓ |
|---|---|---|---|---|---|---|---|
| L2P | 59.22 | 70.01 | 73.16 | 69.24 | 67.91 | 68.69 | 0.35 |
| DualPrompt | 62.64 | 71.76 | 75.43 | 72.63 | 70.62 | 69.77 | 1.53 |
| iDPA (ours) | **66.10** | **74.33** | **78.77** | **77.93** | **74.28** | **70.92** | **-0.03** |

COCO metric (Lin et al., 2014). The average AP across all tasks after learning task $i$ is given by $\mathrm{AP}_i = \frac{1}{i} \sum_{\hat{i}=1}^{i} \mathrm{AP}_{i,\hat{i}}$. After completing all $N$ tasks, the final AP for each task is denoted as $\mathrm{AP}_{N,\_}$, and the final average performance is computed as $\mathrm{FAP} = \mathrm{AP}_N$, which serves as the primary metric for CL performance. To assess historical performance, we calculate $\mathrm{CAP} = \frac{1}{N} \sum_{i=1}^{N} \mathrm{AP}_i$, and the Final Forgetting Percentage as $\mathrm{FFP} = \frac{1}{N-1} \sum_{i=1}^{N-1} (\mathrm{AP}_{i,i} - \mathrm{AP}_{N,i})$, extending the Forgetting Percentage Point (FPP) introduced by CL-DETR (Liu et al., 2023b).

**Comparison Methods.** We compare our iDPA with both non-prompt-based and prompt-based CL methods. For non-prompt-based CL, we select Sequential, ER (Rolnick et al., 2019), WiSE-FT (Wortsman et al., 2022), and ZiRa (Deng et al., 2024). For prompt-based CL, we include L2P (Wang et al., 2022c), DualPrompt (Wang et al., 2022b), CODA (Smith et al., 2023), S-Prompt (Wang et al., 2022a), DIKI (Tang et al., 2025), and NoRGa (Le et al., 2024), which follow a similar task-specific parameter training approach to our iDPA. Note that L2P, DualPrompt, CODA, and NoRGa are originally designed for vision tasks; we extend them to multimodal tasks for fair comparison. For the all-in-one style foundation model, we compared Zero-shot GLIP, SAM2.1 (Ravi et al., 2024), and MedSAM-2 (Zhu et al., 2024). More details on reproduction can be found in the appendix.

**Implementation Details.** We use the GLIP (Liu et al., 2023a) model with Swin-T (Liu et al., 2021), pre-trained

on Object365 (Shao et al., 2019), GoldG (Liu et al., 2023a), and Cap4M (Liu et al., 2023a), as a robust starting point. All experiments employ AdamW (Loshchilov, 2017) with a multistep learning rate scheduler. The learning rate is set to 0.1, and weight decay is set to 1e-4. The experiments run on 4 GPUs with a batch size of 1 per GPU for 5 epochs, with a learning rate decay of 0.1 at epoch 3. All results are averaged over 3 random seeds, with the task order determined by the seed. All comparison methods are re-implemented based on their official implementations. For more details, please refer to the appendix.

### 5.2. Main Results

**Full Data Setting.** Tab. 1 presents the final performance for each task on ODinM-13 under full data training, along with the FAP, CAP, and FFP scores to evaluate the model's final performance, overall performance, and resistance to forgetting. The "Zero-shot" results represent the starting point, derived by leveraging the original GLIP weights for each task. The "FT" results indicate the model's oracle performance, which is achieved by training on a single task and testing on the corresponding task. The "Joint" results represent the model trained on the datasets of all tasks, serving as the upper bound in continual learning.

As indicated by the bold values, iDPA achieves the best final performance on 9 out of 13 tasks compared to other methods. It outperforms the previous prompt-based SOTA method, NoRGa (Le et al., 2024), by 5.44% in FAP, 4.20% in CAP, and reduces FFP by 2.44%. It also outperforms the previous non-prompt-based SOTA method, ER (Rolnick et al., 2019), which requires extra data (10-shot per class) for rehearsal and full model tuning, by 10.37% in FAP, 5.37% in CAP, and reduces FFP by 16.77%. Furthermore, iDPA uses only 1.4% of the trainable parameters and does not require additional data for rehearsal. Compared to ZiRa (Deng et al.,

Table 4: Ablation study of key components in iDPA.

| Method | FAP ↑ | CAP ↑ | FFP ↓ | #Params↓ | #Memory↓ |
|---|---|---|---|---|---|
| Naïve | 44.99 | 49.86 | 4.31 | **1.24M** | 7200M |
| IPG (w/ T) | 48.65 | 52.69 | 3.71 | 3.21M | 7285M |
| DPA | 47.10 | 51.04 | 3.73 | 1.37M | **6496M** |
| IPG (w/o T) + DPA | 50.17 | 53.89 | 2.54 | 3.22M | 6590M |
| IPG (w/ T) + DPA | **50.28** | **54.10** | **2.48** | 3.22M | 6590M |

Table 5: Comparison of iDPA at different positions.

| $\Phi_v$ | $\Phi_t$ | $\Phi_f$ | FAP ↑ | CAP ↑ | FFP ↓ | #Params ↓ | #Memory ↓ | #Time ↓ |
|---|---|---|---|---|---|---|---|---|
| ✓ | ✗ | ✗ | 47.44 | 51.78 | 2.91 | 4.36M | 7622M | 7h40m |
| ✗ | ✓ | ✗ | 38.44 | 44.89 | 7.53 | 6.89M | 7177M | 7h36m |
| ✓ | ✓ | ✗ | 48.41 | 53.56 | 4.38 | 11.26M | 8295M | 9h10m |
| ✗ | ✗ | ✓ | 50.28 | 54.10 | **2.48** | **3.22M** | **6590M** | **6h28m** |
| ✓ | ✓ | ✓ | **51.56** | **56.25** | 3.17 | 14.48M | 8945M | 9h32m |

**Notes: FAP** (%): Final Average AP. **CAP** (%): Cumulative Average AP. **FFP** (%): Final Forgetting Percentage of old tasks. **#Params**: The number of trainable parameters used during training. **#Memory**: Memory usage during training with 1024 × 1024 input and batch size 1. **#Time**: Training time on ODinM-13 using 1 GPU, batch size 1, for 5 epochs. **w/ T**: Indicates the presence of weight transfer between tasks. **w/o T**: Indicates the absence of weight transfer between tasks.

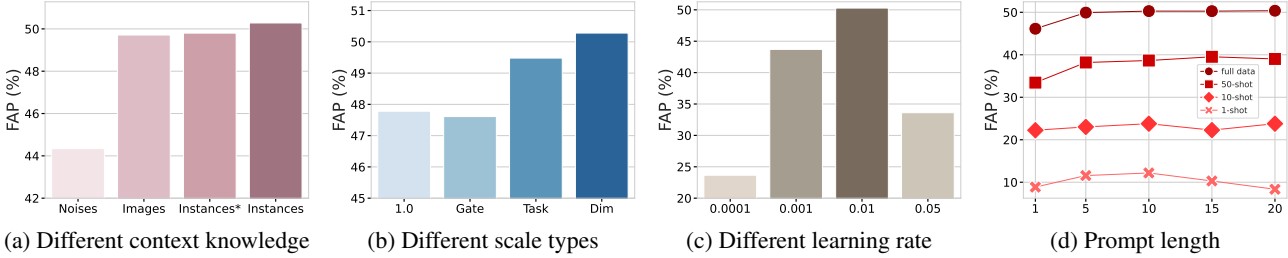

(a) Different context knowledge  (b) Different scale types  (c) Different learning rate  (d) Prompt length

Figure 5: Impact of context knowledge, scale types, learning rate, and prompt length on model performance.

Table 6: Comparison of iDPA with varying layers in $\Phi_f$.

| Layers | FAP ↑ | $FAP^{50}$ ↑ | $FAP^{75}$ ↑ | #Params ↓ | #Memory ↓ |
|---|---|---|---|---|---|
| 1 | 45.53 | 75.07 | 46.78 | **0.54M** | **6054M** |
| 2 | 48.16 | 77.72 | 50.89 | 1.07M | 6162M |
| 4 | 49.94 | 79.12 | 53.34 | 2.15M | 6371M |
| 6 (all) | 50.28 | 79.47 | 54.29 | 3.22M | 6590M |

Table 7: Impact of scaling factor $\gamma$ in Eq. 7 on performance.

| $\gamma$ | FAP ↑ | AP↑ | FFP ↓ |
|---|---|---|---|
| 1.00 | 50.07 | 54.01 | 2.57 |
| 1.30 | **50.28** | **54.10** | **2.48** |
| 1.50 | 49.99 | 53.40 | 2.77 |

Table 8: Comparison between the Naïve baseline and iDPA on ($\Phi_v + \Phi_t$).

| ($\Phi_v + \Phi_t$) | FAP ↑ | CAP ↑ | FFP ↓ |
|---|---|---|---|
| Naïve | 41.72 | 47.30 | 8.57 |
| iDPA (ours) | 48.41 | 53.56 | 4.38 |
| $\Delta$ | 6.69 | 6.26 | 4.19 |

2024), which is designed for incremental VLOD learning, iDPA surpasses it by 46.62% in FAP, 37.73% in CAP, and reduces FFP by 47.19%. This is due to the substantial gap between the medical and natural domains, along with the large differences across modalities, organs, and categories, which makes it difficult for ZiRa to regularize and reconfigure parameters to learn a shared space. Additionally, compared to the upper bound, iDPA is only 4.39% lower in FAP. These results demonstrate the effectiveness of using a prompt-based CL method for the comprehensive MOD task. Our method maintains excellent performance while ensuring data independence, parameter efficiency, model resistance to forgetting, scalability, and flexibility.

**Few-Shot Data Setting.** To simulate the challenging real-world scenario of limited data annotation in clinical settings, we also conduct experiments in the few-shot setting on ODinM-13, as shown in Tab. 2. In the 1-shot setting, our iDPA outperforms the best alternative by 4.38% in FAP, and 4.07% in CAP, with only a 0.44% increase in FFP. In the 10-shot setting, iDPA outperforms the best result by 12.88% in FAP, 8.65% in CAP, and exhibits a minimal 0.39% increase in FFP. In the 50-shot setting, iDPA outperforms the best result by 4.59% in FAP, 2.76% in CAP, and reduces FFP by 0.44%. These results highlight the strong knowledge

transfer capability of our approach, which, by decoupling instance-level knowledge and leveraging DPA, greatly improves model performance and efficiency, particularly in data-scarce environments.

**Domain Continual Setting.** iDPA achieves the best performance across all four polyp datasets under the continual domain setting, outperforming L2P and DualPrompt in FAP, CAP, and FFP. These results demonstrate the superior generalizability and stability of iDPA in handling domain shifts.

**Visualization.** Fig. 4 shows the AP variation of different CL methods on ODinM-13. Our method outperforms existing ones throughout the incremental learning process, not just at the end. More qualitative results are provided in Fig. 7 in the appendix showing that iDPA produces more accurate bounding boxes with higher confidence for various MOD tasks

than Zero-shot and the L2P method (Wang et al., 2022c), using enhanced knowledge transfer.

### 5.3. Ablation Study

We conduct ablation studies of the proposed modules in iDPA on ODinM-13 under the full data setting, as shown in Tab. 4. Compared to the Naïve prompt method, adding the IPG module improves FAP by 3.66%, CAP by 2.83%, and reduces FFP by 0.60%. Adding the DPA module increases FAP by 2.11%, CAP by 1.18%, and reduces FFP by 0.58%. Moreover, DPA reduces both gradient backpropagation computation and memory usage, while introducing only a minimal number of additional parameters. When both modules are combined, FAP increases by 5.29%, CAP by 4.24%, and FFP decreases by 1.83%. These results demonstrate that decoupling instance knowledge from images effectively enhances object recognition and localization. By decoupling PA, DPA enables more efficient learning and better injection of prompt knowledge. Furthermore, since the two modules are orthogonal, combining them improves the model's ability to complete the IMOD task. Additionally, we investigate knowledge transfer across medical tasks. When IPG is not used for weight transfer, performance slightly decreases. However, despite the substantial differences between medical tasks, knowledge sharing still occurs. This is especially evident when reducing the number of training epochs, where weight transfer significantly boosts learning efficiency. For further details, please refer to the appendix.

### 5.4. Empirical Analysis

**Impact of Knowledge Injection Position.** As shown in Tab. 5, we compare different positions for prompt knowledge injection in VLOD models. The Fusion Encoder achieves the best balance between performance and cost.

**Impact of Context Knowledge.** In Fig. 5a, we compare four types of context knowledge: Gaussian noise, image knowledge, instance-level knowledge from different layers (denoted as 'Instances*'), and instance-level knowledge from the corresponding layer (denoted as 'Instances'). Our experiments demonstrate that context-aware knowledge enhances the IMOD task, with the best performance achieved by instance knowledge from the corresponding layer.

**Impact of Scale $\lambda$ in DPA.** We test four types of scale $\lambda$ in DPA: constant (1.0), gate mechanism, task-level ($\lambda \in \mathbb{R}^{1 \times 1}$), and dim-level ($\lambda \in \mathbb{R}^{1 \times d}$). As shown in Fig. 5b, dim-level $\lambda$ yields the best performance.

**Impact of Learning Rate.** A grid search over the range [1e-5, 0.1] reveals that a learning rate of 0.01 provides the best performance. Results for iDPA at different learning rates are shown in Fig. 5c.

**Impact of Prompt Length.** We compare prompt lengths in full data and few-shot settings (Fig. 5d). A prompt length of 10 offers balanced performance and is chosen as the default.

**Impact of $X$-Attn Layer Count.** We conduct experiments with different numbers of $X$-Attn layers (1, 2, 4, and 6) and find that incorporating all layers achieves the best performance, as shown in Tab. 6.

**Impact of scaling factor $\gamma$ in Eq. 7.** As shown in Tab. 7, the model achieves the best performance when the scaling factor $\gamma$ is set to 1.30, yielding the highest FAP and AP, and the lowest FFP. This suggests that moderately enlarging the RoI region helps capture more useful contextual information.

**Fairness of Naïve Baseline Comparison.** As shown in Tab. 8, iDPA outperforms the Naïve baseline when prompts are injected into both $\Phi_v$ and $\Phi_t$, achieving gains of 6.69 FAP and 6.26 CAP, and reducing FFP by 4.19, demonstrating the effectiveness of our design.

## 6. Conclusion

This study presents iDPA for IMOD learning without catastrophic forgetting. iDPA efficiently generates and injects instance-level knowledge, reducing computational complexity and memory usage. It decouples target instance features and employs a continual concept perception mechanism to create and integrate concept prompts into the multimodal fusion process. Additionally, iDPA refines prompt attention into three key interaction steps, focusing on continual learning for efficient knowledge injection while preserving the original knowledge. For the evaluation, we introduce a new IMOD benchmark, ODinM-13, with 13 MOD datasets. Experiments show that iDPA outperforms previous SOTA methods in both full-data and few-shot settings. Our analysis also demonstrates that our method can be more efficient and memory-friendly compared to previous CL methods.

## Impact Statement

This paper presents work whose goal is to advance the field of Machine Learning. There are many potential societal consequences of our work, none of which we feel must be specifically highlighted here.

## Acknowledgments

The authors would like to thank the GLIP team and the developers of other continual learning methods for making their code and models publicly available, which greatly facilitated this work. We also acknowledge the contributions of individuals and organizations who have shared open-source medical imaging datasets. We are grateful to the anonymous reviewers and program chairs for their valuable and constructive feedback.

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

# A. Theoretical Analysis

This section presents a theoretical analysis demonstrating the superior efficiency of DPA over traditional prompt learning.

## A.1. Overall Analysis

$$
\begin{aligned}
f_1 = \mathrm{Concat}[&(1 - \lambda(p_t))\mathrm{Attn}_{v \to t}(f_v, p_t) + \lambda(p_t)\mathrm{Attn}_{v \to t}(p_v, p_t); \\
&(1 - \lambda(f_t))\mathrm{Attn}_{v \to t}(f_v, f_t) + \lambda(f_t)\mathrm{Attn}_{v \to t}(p_v, f_t)], \\
= \mathrm{Concat}[&A, B]
\end{aligned} \tag{14}
$$

where $p_{v,t} \in \mathbb{R}^{l \times d}$ represent the vision and text prompts, and $f_v, f_t$ are the visual and textual features before being fed into $\mathrm{Attn}_{v \to t}$.

$$
f_2 = (1 - \lambda(f_t))\mathrm{Attn}_{v \to t}(f_v, f_t) + \lambda(f_t)\mathrm{Attn}_{v \to t}(p_v, f_t) = B, \tag{15}
$$

If $\mathrm{Attn}_{v \to t}(\cdot, \cdot) \in \mathbb{R}^{L \times d}$, then the two terms $A(p_t)$ and $A(f_t)$ each belong to $\mathbb{R}^{L \times d}$ and $f_1 \in \mathbb{R}^{L \times 2d}$. In contrast, $f_2 \in \mathbb{R}^{L \times d}$.

$$
f_2 = \mathrm{Attn}_{v \to t}(f_v, f_t) + \lambda(f_t)\Delta, \tag{16}
$$

where we have defined $\Delta = \mathrm{Attn}_{v \to t}(p_v, f_t) - \mathrm{Attn}_{v \to t}(f_v, f_t)$.

$$
\frac{\partial f_2}{\partial \theta} = \frac{\partial \mathrm{Attn}_{v \to t}(f_v, f_t)}{\partial \theta} + \lambda(f_t)\frac{\partial \Delta}{\partial \theta}. \tag{17}
$$

This indicates that $f_2$ has a lower-dimensional structure, residual components, and a more direct gradient flow.

## A.2. Computation Cost

**Lemma A.1.** *$f_2$ is computational light than $f_1$*

*Proof.* The overall $f_1$ is $f_1 = \mathrm{Concat}[A; B]$. Howeveer, $f_2$ uses only the $B$ branch,$f_2 = B$. The computation cost of branch $A$ is roughly $\mathcal{O}_A = \mathcal{O}(f_v, p_t) + \mathcal{O}(p_v, p_t) = \mathcal{O}(A + B)$ for B $\mathcal{O}_{f_2} = \mathcal{O}(B)$. $\mathcal{O}_{f_1} > \mathcal{O}_{f_2}$. $\qquad\square$

## A.3. convergence benefit analysis

**Lemma A.2.** *Let $f_1$ and $f_2$ be our models. Assume both models have converged to any local minima and there be an optimal representation $f^* = B$. Suppose further output of $f_1$ is locally linear around the optimum, then $f_2$ achieves the same performance as $f_1$ at the local convergence.*

*Proof.* The loss function $\mathcal{L}(f_{\mathrm{out}})$, For $f$, the output $y_1 = h(\mathrm{Concat}[A; B])$, and $y_2 = h(B)$. At convergence: $\nabla \mathcal{L}(f_1) = \mathbf{0}$ and $\nabla \mathcal{L}(f_2) = \mathbf{0}$. There exists an optimal representation $f^*$ such that it is sufficient to have $f^* = B$.

Assume $h$ is locally linear at the optima, then there exists a matrix $M$ such that

$$
h\Big(\mathrm{Concat}[A; B]\Big) = M \begin{pmatrix} A \\ B \end{pmatrix} = M_1 A + M_2 B. \tag{18}
$$

Under convergence, $M_1 A = \mathbf{0}$, thus

$$
h\Big(\mathrm{Concat}[A; B]\Big) = M_2 B = h'(B). \tag{19}
$$

$$
y_1 = h\Big(\mathrm{Concat}[A; B]\Big) = h'\Big(B\Big) = y_2. \tag{20}
$$

Thus $f_2$ performs as well as $f_1$ after convergence to local minima. $\qquad\square$

## B. More Implementation Details

**Benchmark.** We collected 13 public datasets from the internet: DFUC-2020 (DFUC) (Cassidy et al., 2021), Kvasir (Jha et al., 2020), OpticNerv (OpticN) [1], BCCD [2], CPM-17 (Vu et al., 2019), Breast Cancer (BreastC) [3], TBX11K (Liu et al., 2020), Kidney Tumor (KidneyT) [4], Luna16 (Setio et al., 2017), ADNI (Boccardi et al., 2015), Meningioma (Meneng) [5], Breast Tumor (BreastT) [6], and TN3K (Gong et al., 2021). Among them, OpticN, BCCD, BreastC, KidneyT, Meneng, and BreastT are from the Roboflow [7] website. These datasets include eight different modalities: Photography, Endoscopy, Light Microscopy, Histopathology, X-ray, CT, MRI, and Ultrasound, covering nine different organs: Foot, Colorectal, Nerve, Blood/Cell, Lung, Brain, Breast, Kidney, and Thyroid. The random seed used for few-shot data generation is kept consistent with the one used during training. $k$-shot means ensuring that each class in the current dataset contains at least $k$ instances. Three different orders were used during training. The dataset order and corresponding random seed are shown in the Tab. 9.

Table 9: Task order under different random seeds. The table shows the dataset sequences used during training for three different random seeds (0, 5, and 10).

| Order | 1 | 2 | 3 | 4 | 5 | 6 | 7 | 8 | 9 | 10 | 11 | 12 | 13 |
|---|---|---|---|---|---|---|---|---|---|---|---|---|---|
| Seed 0 | DFUC | Kvasir | OpticN | BCCD | CPM-17 | BreastC | TBX11K | KidneyT | Luna16 | ADNI | Meneng | BreastT | TN3k |
| Seed 5 | OpticN | BCCD | BreastT | Meneng | TN3k | Kvasir | TBX11K | KidneyT | DFUC | Luna16 | BreastC | CPM-17 | ADNI |
| Seed 10 | CPM-17 | BreastC | Luna16 | Kvasir | OpticN | Meneng | TN3k | BCCD | BreastT | KidneyT | TBX11K | DFUC | ADNI |

**Implementation.** The proposed method is implemented in Python using the PyTorch library and runs on a PC. The code is based on the official GLIP (Li et al., 2022) implementation [8], and its environment requirements remain unchanged. For full-data training, we use four NVIDIA 3090 GPUs with a batch size of 4, while for few-shot training, we use a single NVIDIA 3090 GPU with a batch size of 1. Unless otherwise specified, all experiments are trained for 5 epochs, with the learning rate reduced by a factor of 0.1 after 3 epochs. For all prompt-based CL methods (Wang et al., 2022c;b; Smith et al., 2023; Wang et al., 2022a; Tang et al., 2025; Le et al., 2024), the initial learning rate is set to 1e-2, whereas ZiRa (Deng et al., 2024) uses an initial learning rate of 1e-3. Standard fine-tuning (FT), joint training, sequential training, WiSE-FT (Wortsman et al., 2022), and experience replay (ER) (Rolnick et al., 2019) use an initial learning rate of 1e-4. The learning rates are determined via grid search within the range of [1e-5, 0.1]. To ensure reproducibility, all experiments are conducted with three different random seeds (0, 5, 10), and the dataset order is adjusted accordingly. The final results are reported as the average over three runs.

**Reproduction.** We reproduce other prompt-based methods on GLIP by prompting all layers of both the vision and text encoders, whereas the original papers typically use only the embedding layer or a few initial layers (e.g., the first five layers). This discrepancy may lead to suboptimal performance on the IMOD task. The vision backbone is used as the query function, and the mean feature representation from its last layer is utilized to identify the task ID. For L2P (Wang et al., 2022c), we set the prompt length to 5. During inference, the top-5 prompts are selected from the prompt pool, following the official implementation. In the original L2P paper, updated prompts are selected via a key-matching mechanism during training, with diversity maintained using a frequency-based weighting technique. However, in the official code repository, specific prompts are masked for different tasks. We follow the implementation provided in the official code. For DualPrompt (Wang et al., 2022b), we set the prompt length to 10 for both key and value prompts. Two layers are designated as G(eneral)-Prompt, while the remaining 10 layers serve as E(xpert)-Prompt. For CODA (Smith et al., 2023), we set the prompt length to 8 and

---

[1] https://universe.roboflow.com/neurosurgery/optic-nerv
[2] https://public.roboflow.com/object-detection/bccd
[3] https://universe.roboflow.com/tez-m06pk/breast-cancer-tbwa9
[4] https://universe.roboflow.com/east-delta-university-rpdgs/kidney_tumor-tke8k
[5] https://universe.roboflow.com/mem-g72lg/menengioma
[6] https://universe.roboflow.com/qidiliu/breast-tumor-detection-nsikz
[7] https://roboflow.com/
[8] https://github.com/microsoft/GLIP

additionally add an extra key to learn the task identity, following the official implementation. For S-Prompt (Wang et al., 2022a), DIKI (Tang et al., 2025), and NoRGa (Le et al., 2024), we set the prompt length to 10. S-Prompt employs K-Means to generate 5 prototypes for task identification. All reproductions adhere to the implementation in the official code.

## C. Additional Results

Tab. 10 compares the impact of weight transfer between tasks when training for 3 epochs. The results show that enabling weight transfer improves continual learning performance under limited training time. Tab. 11 presents a detailed comparison of iDPA with different knowledge injection positions. The best performance is observed when knowledge is injected simultaneously into the vision, text, and fusion encoders. However, this setting leads to a higher forgetting rate compared to injecting knowledge only in the fusion encoder. Fig. 6 visualizes the performance dynamics across different knowledge injection positions. Tab. 12, Tab. 13, and Tab. 14 report the performance of iDPA compared with other continual learning methods under 1-shot, 10-shot, and 50-shot settings, respectively. Tab. 15 shows that iDPA achieves the lowest parameter count (3.34M), reduced FLOPs, and significantly lower memory consumption and training time compared to all baselines. While maintaining competitive inference speed (5.93 FPS), it offers the best overall efficiency among the evaluated continual learning methods. Tab. 16 demonstrates that iDPA consistently improves performance across 13 datasets under 1-shot, 5-shot, and 10-shot settings, with low variance indicating strong stability and generalization. Fig. 7 provides qualitative comparisons between iDPA, Ground Truth, Zero-shot, and L2P (Wang et al., 2022c), the pioneering prompt-based continual learning method. iDPA shows superior localization accuracy, more precise classification, and higher confidence scores.

Table 10: Performance of using weight transfer over 3 epochs. **FAP** (%): Final Average AP. **CAP** (%): Cumulative Average AP.

| Transfer | DFUC | Kvasir | OpticN | BCCD | CPM-17 | BreastC | TBX11K | KidneyT | Luna16 | ADNI | Meneng | BreastT | TN3k | FAP ↑ | CAP ↑ | FFP ↓ |
|---|---|---|---|---|---|---|---|---|---|---|---|---|---|---|---|---|
| W/O | 46.86 | 73.69 | 63.61 | 60.25 | 29.88 | 50.68 | 30.59 | 57.88 | 30.58 | 42.80 | 50.84 | 19.23 | 51.28 | 46.78 | 51.87 | 3.98 |
| W | 46.34 | 71.05 | 66.86 | 60.33 | 30.65 | 49.16 | 29.39 | 63.78 | 28.16 | 39.99 | 56.97 | 27.10 | 52.36 | 47.86 | 52.06 | 2.47 |

Table 11: Comparison of performance with different knowledge injection positions. **FAP** (%): Final Average AP. **CAP** (%): Cumulative Average AP. **FFP** (%): Final Forgetting Percentage of old tasks.

| $\Phi_v$ | $\Phi_t$ | $\Phi_f$ | DFUC | Kvasir | OpticN | BCCD | CPM-17 | BreastC | TBX11K | KidneyT | Luna16 | ADNI | Meneng | BreastT | TN3k | FAP ↑ | CAP ↑ | FFP ↓ |
|---|---|---|---|---|---|---|---|---|---|---|---|---|---|---|---|---|---|---|
| ✓ | ✗ | ✗ | 42.28 | 72.53 | 66.30 | 58.94 | 27.08 | 49.73 | 30.25 | 62.38 | 27.51 | 38.23 | 56.00 | 32.98 | 52.46 | 47.44 | 51.78 | 2.91 |
| ✗ | ✓ | ✗ | 40.92 | 70.24 | 29.61 | 47.24 | 26.08 | 42.09 | 20.45 | 55.47 | 23.58 | 22.92 | 56.93 | 26.13 | 38.10 | 38.44 | 44.89 | 7.53 |
| ✓ | ✓ | ✗ | 46.57 | 73.05 | 59.51 | 59.82 | 34.62 | 47.35 | 32.15 | 62.24 | 29.69 | 41.98 | 58.83 | 29.51 | 54.01 | 48.41 | 53.56 | 4.38 |
| ✗ | ✗ | ✓ | 47.09 | 73.76 | **66.85** | 60.29 | **36.54** | 50.98 | 32.69 | 64.98 | 31.15 | 44.42 | 57.20 | 34.65 | 53.03 | 50.28 | 54.10 | **2.48** |
| ✓ | ✓ | ✓ | **49.40** | **76.04** | 66.40 | **60.91** | 33.15 | **53.12** | **36.43** | **65.73** | **32.19** | **45.94** | **58.15** | **35.67** | **57.19** | **51.56** | **56.25** | 3.17 |

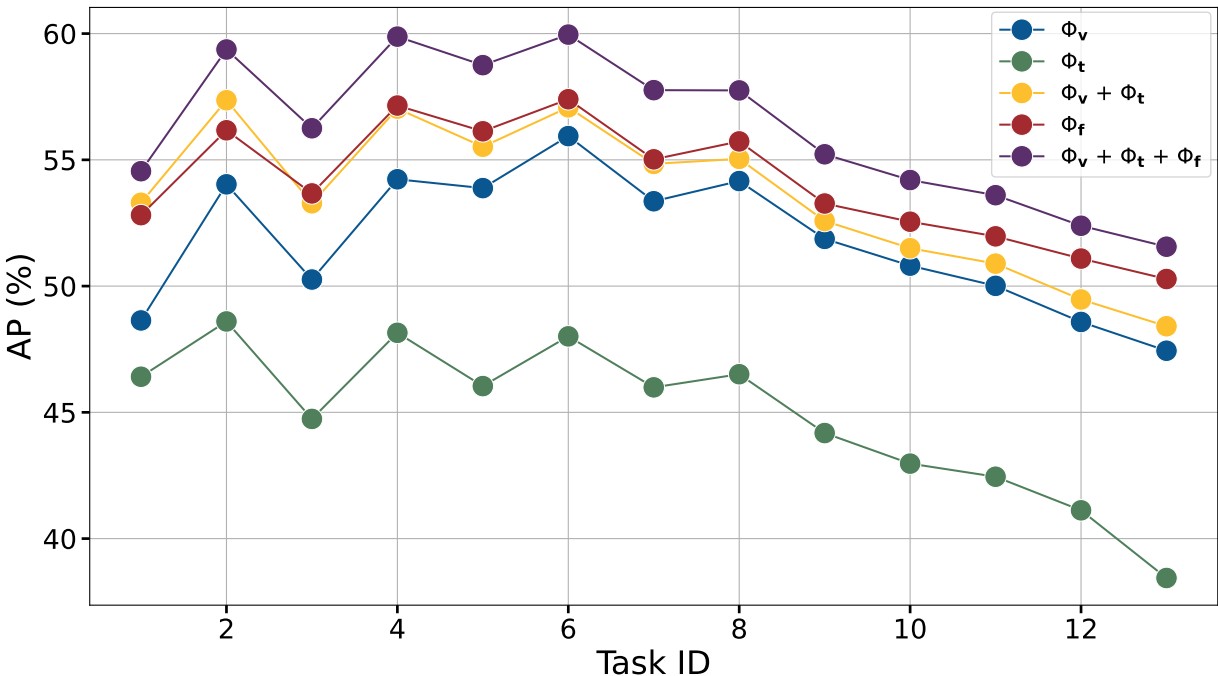

Figure 6: Performance variation of iDPA in different location.

Table 12: Performance of various continual learning methods on the ODinM-13 benchmark under the 1-shot setting. **FAP** (%): Final Average AP. **CAP** (%): Cumulative Average AP. **FFP** (%): Final Forgetting Percentage of old tasks.

| Methods | DFUC | Kvasir | OpticN | BCCD | CPM-17 | BreastC | TBX11K | KidneyT | Luna16 | ADNI | Meneng | BreastT | TN3k | FAP↑ | CAP↑ | FFP↓ |
|---|---|---|---|---|---|---|---|---|---|---|---|---|---|---|---|---|
| Joint (Upper) | 7.99 | 51.35 | 24.88 | 33.54 | 36.28 | 9.18 | 1.94 | 22.43 | 4.20 | 1.54 | 46.18 | 12.35 | 8.48 | 20.03 | - | - |
| **Non-Prompt-based CL** | | | | | | | | | | | | | | | | |
| Sequential | 0.00 | 0.00 | 0.00 | 0.00 | 8.70 | 0.00 | 0.00 | 0.00 | 0.06 | 0.00 | 1.87 | 0.00 | 5.54 | 1.24 | 11.49 | 23.43 |
| WiSE-FT | 4.63 | 26.75 | 0.32 | 9.04 | 9.75 | 0.32 | 0.08 | 6.33 | 0.03 | 0.00 | 11.37 | **7.89** | 3.52 | 6.16 | 14.62 | 9.25 |
| ZiRa | 6.17 | 26.87 | 2.40 | 3.62 | 12.34 | **7.66** | 0.56 | 4.57 | 0.00 | 0.01 | 19.42 | 4.85 | 2.23 | 6.98 | 13.59 | 11.68 |
| **Prompt-based CL** | | | | | | | | | | | | | | | | |
| L2P | 1.56 | 2.16 | 0.00 | 10.46 | 12.45 | 1.14 | 0.21 | 3.69 | 0.25 | 0.00 | 7.91 | 0.43 | 1.99 | 3.25 | 7.18 | **3.14** |
| DualPrompt | 6.20 | 32.05 | 1.01 | 9.64 | 11.00 | 0.48 | 0.00 | 3.61 | 0.23 | 0.10 | **19.63** | 6.20 | 5.49 | 7.36 | 13.30 | 7.93 |
| S-Prompt | 2.24 | 7.94 | 3.55 | 6.68 | 8.72 | 2.35 | 0.42 | 3.25 | 0.02 | 0.00 | 5.69 | 1.00 | 1.53 | 3.34 | 8.90 | 5.66 |
| CODA | 0.76 | 2.69 | 7.19 | **24.02** | 20.42 | 0.34 | 0.92 | 0.69 | **1.03** | 0.13 | 14.95 | 5.73 | **10.68** | 6.89 | 13.96 | 4.57 |
| DIKI | 0.54 | 11.07 | 3.44 | 16.95 | 16.22 | 2.41 | 0.62 | **8.71** | 0.01 | 0.01 | 6.89 | 3.54 | 3.30 | 5.67 | 13.09 | 3.88 |
| NoRGa | 1.00 | 5.73 | 13.86 | 17.62 | 11.24 | 0.64 | 0.56 | 7.31 | 0.09 | 0.01 | 8.50 | 8.13 | 3.57 | 6.02 | 11.09 | 5.17 |
| iDPA(Ours) | **6.66** | **43.04** | **14.62** | 20.55 | **31.13** | 5.33 | **2.15** | 7.38 | 0.30 | **0.39** | 17.34 | 6.17 | 3.45 | **12.19** | **18.03** | 3.58 |

Table 13: Performance of various continual learning methods on the ODinM-13 benchmark under the 10-shot setting. **FAP** (%): Final Average AP. **CAP** (%): Cumulative Average AP. **FFP** (%): Final Forgetting Percentage of old tasks.

| Methods | DFUC | Kvasir | OpticN | BCCD | CPM-17 | BreastC | TBX11K | KidneyT | Luna16 | ADNI | Meneng | BreastT | TN3k | FAP↑ | CAP↑ | FFP↓ |
|---|---|---|---|---|---|---|---|---|---|---|---|---|---|---|---|---|
| Joint (Upper) | 32.60 | 60.99 | 56.07 | 56.14 | 41.01 | 33.29 | 10.49 | 44.17 | 13.38 | 10.55 | 67.57 | 20.77 | 29.76 | 36.68 | - | - |
| **Non-Prompt-based CL** | | | | | | | | | | | | | | | | |
| Sequential | 3.84 | 0.00 | 0.00 | 0.00 | 0.36 | 0.00 | 0.00 | 0.00 | 0.00 | 8.19 | 0.00 | 1.92 | 7.33 | 1.66 | 13.67 | 36.51 |
| WiSE-FT | 7.13 | 41.13 | 1.45 | 12.68 | 16.91 | 1.91 | 0.13 | 8.98 | 0.06 | 0.04 | 16.31 | 10.68 | 5.69 | 9.47 | 21.03 | 12.98 |
| ZiRa | 6.07 | 30.71 | 1.32 | 11.14 | 16.38 | 9.18 | 0.53 | 7.00 | 0.44 | 0.57 | 40.70 | 8.74 | 8.96 | 10.90 | 16.19 | 15.12 |
| **Prompt-based CL** | | | | | | | | | | | | | | | | |
| L2P | 2.92 | 16.65 | 2.46 | 17.90 | 11.04 | 4.08 | 0.09 | 4.34 | 0.02 | 0.02 | 3.86 | 1.03 | 2.72 | 5.16 | 9.63 | **4.48** |
| DualPrompt | 1.95 | 6.19 | 0.06 | 14.48 | 7.17 | 0.91 | 0.06 | 0.76 | 0.06 | 0.06 | 36.14 | 6.63 | 2.91 | 5.95 | 13.99 | 11.34 |
| S-Prompt | 8.20 | 16.33 | 9.13 | 9.12 | 10.23 | 5.53 | 0.43 | 8.01 | 0.25 | 0.13 | 37.67 | 1.13 | 2.95 | 8.39 | 12.98 | 4.36 |
| CODA | 5.23 | 0.23 | 1.33 | 25.57 | 3.34 | 0.28 | 0.22 | 2.10 | 0.01 | 0.01 | 7.78 | 4.49 | 6.78 | 4.41 | 14.20 | 12.60 |
| DIKI | 0.97 | 19.66 | 0.25 | 27.92 | 5.83 | 1.27 | 1.51 | 9.31 | 0.08 | 0.01 | 6.11 | 7.20 | 3.86 | 6.46 | 15.34 | 9.78 |
| NoRGa | 2.60 | 11.19 | 2.96 | 24.84 | 3.01 | 1.42 | 0.18 | 0.69 | 0.01 | 0.01 | 14.49 | 6.51 | 1.80 | 5.36 | 11.90 | 8.14 |
| iDPA(Ours) | **21.37** | **50.20** | **29.20** | **39.11** | **38.33** | **19.65** | **6.03** | **27.06** | **6.23** | **3.15** | **39.42** | **15.34** | **14.02** | **23.78** | **29.68** | 4.75 |

Table 14: Performance of various continual learning methods on the ODinM-13 benchmark under the 50-shot setting. **FAP** (%): Final Average AP. **CAP** (%): Cumulative Average AP. **FFP** (%): Final Forgetting Percentage of old tasks.

| Methods | DFUC | Kvasir | OpticN | BCCD | CPM-17 | BreastC | TBX11K | KidneyT | Luna16 | ADNI | Meneng | BreastT | TN3k | FAP↑ | CAP↑ | FFP↓ |
|---|---|---|---|---|---|---|---|---|---|---|---|---|---|---|---|---|
| Joint (Upper) | 39.56 | 66.52 | 64.41 | 59.26 | 41.78 | 43.82 | 26.48 | 52.90 | 22.61 | 66.40 | 72.93 | 25.61 | 47.92 | 48.48 | - | - |
| **Non-Prompt-based CL** | | | | | | | | | | | | | | | | |
| Sequential | 0.00 | 0.00 | 0.00 | 0.00 | 5.51 | 0.00 | 0.00 | 0.00 | 0.00 | 22.19 | 0.00 | 0.00 | 16.20 | 3.38 | 14.95 | 46.86 |
| WiSE-FT | 8.18 | 43.77 | 1.29 | 13.05 | 14.99 | 2.93 | 0.11 | 10.59 | 0.08 | 0.05 | 21.55 | 10.08 | 6.44 | 10.24 | 23.34 | 16.03 |
| ZiRa | 2.07 | 5.58 | 0.39 | 5.45 | 4.89 | 3.56 | 0.23 | 0.78 | 0.45 | 3.03 | 4.85 | 5.51 | 8.58 | 3.49 | 15.78 | 37.28 |
| **Prompt-based CL** | | | | | | | | | | | | | | | | |
| L2P | 29.13 | 54.05 | 34.74 | 38.00 | 35.61 | 38.23 | 6.41 | 34.59 | 13.33 | 2.74 | 46.20 | 10.18 | 19.67 | 27.91 | 35.20 | 5.66 |
| DualPrompt | 24.84 | 27.49 | 25.44 | 47.18 | 27.36 | 29.17 | 3.33 | 21.87 | 13.09 | 6.57 | 16.32 | 9.28 | 13.82 | 20.44 | 34.94 | 16.91 |
| S-Prompt | 23.79 | 39.45 | 33.67 | 16.90 | 8.74 | 29.77 | 6.00 | 28.24 | 8.38 | 4.78 | **61.73** | 13.08 | 21.40 | 22.76 | 30.86 | 7.92 |
| CODA | 27.42 | 65.27 | 31.74 | 40.66 | 25.85 | 36.81 | **17.28** | 43.72 | 16.37 | 6.93 | 54.25 | **20.55** | 25.91 | 31.75 | 40.86 | 4.37 |
| DIKI | 34.26 | **68.82** | 34.05 | 53.26 | 33.02 | 38.06 | 13.93 | 50.27 | 20.74 | 5.42 | 48.16 | 8.17 | 34.70 | 34.06 | 42.27 | 5.85 |
| NoRGa | 30.69 | 58.17 | 39.23 | 53.81 | 28.12 | **38.82** | 15.31 | 46.16 | 15.21 | 6.76 | 44.04 | 18.96 | 21.87 | 32.09 | 39.12 | 5.70 |
| iDPA(Ours) | **34.79** | 59.03 | **52.64** | **58.12** | **39.33** | 37.35 | 14.78 | **52.77** | **22.70** | **24.55** | 56.32 | 10.99 | **39.10** | **38.65** | **45.03** | **3.93** |

Table 15: Efficiency comparison of different continual learning methods. Metrics include the number of parameters, floating point operations (FLOPs), memory consumption, total training time, and inference speed. Our method achieves the best trade-off with the lowest parameter count and competitive inference performance.

| Methods | #Params↓ | #FLOPs↓ | #Memory↓ | #Time↓ | Inference Speed↑ |
|---|---|---|---|---|---|
| Joint (Upper) | 231.76M | 488.03 GMac | 13129M | 9h55min | 6.18 FPS |
| Sequential | 231.76M | 488.03 GMac | 13129M | 9h55min | 6.18 FPS |
| WiSE-FT | 231.76M | 488.03 GMac | 13129M | 9h55min | 6.18 FPS |
| ER | 231.76M | 488.03 GMac | 13129M | 11h15min | 6.18 FPS |
| ZiRa | 10.23M | 490.15 GMac | 8377M | 6h25min | 6.11 FPS |
| L2P | 6.97M | 601.50 GMac | 10288M | 7h50min | 5.08 FPS |
| DualPrompt | 4.83M | 583.82 GMac | 9417M | 7h36min | 5.25 FPS |
| S-Prompt | 2.73M | 590.89 GMac | 5366M | 8h24min | 5.13 FPS |
| CODA-Prompt | 10.97M | 583.82 GMac | 9803M | 9h03min | 5.26 FPS |
| DIKI | 8.76M | 583.82 GMac | 9754M | 7h49min | 5.16 FPS |
| NoRGa | 8.76M | 583.82 GMac | 9963M | 8h07min | 5.17 FPS |
| iDPA(Ours) | 3.34M | 506.00/501.00 GMac | 6590M | 5h46min | 5.93 FPS |

Table 16: Mean performance across 13 datasets under 1-shot, 5-shot, and 10-shot settings. $\sigma$ denotes the average improvement of our iDPA method over the baseline. Variance results are included to illustrate the stability across multiple runs.

| Methods | DFUC | Kvasir | OpticN | BCCD | CPM-17 | BreastC | TBX11K | KidneyT | Luna16 | ADNI | Meneng | BreastT | TN3k |
|---|---|---|---|---|---|---|---|---|---|---|---|---|---|
| 1 | 6.66 | 43.04 | 14.62 | 20.55 | 31.13 | 5.33 | 2.15 | 7.38 | 0.30 | 0.39 | 17.34 | 6.17 | 3.45 |
| $\sigma$ | 2.55 | 0.95 | 4.89 | 3.86 | 1.71 | 0.93 | 0.00 | 1.55 | 0.14 | 0.03 | 2.18 | 2.46 | 1.31 |
| 5 | 21.37 | 50.20 | 29.20 | 39.11 | 38.33 | 19.65 | 6.03 | 27.06 | 6.23 | 3.15 | 39.42 | 15.34 | 14.02 |
| $\sigma$ | 1.62 | 0.86 | 3.70 | 3.24 | 2.63 | 1.24 | 0.01 | 0.02 | 1.68 | 1.26 | 1.34 | 1.58 | 3.80 |
| 10 | 34.79 | 59.03 | 52.64 | 58.12 | 39.33 | 37.35 | 14.78 | 52.77 | 22.70 | 24.55 | 56.32 | 10.99 | 39.10 |
| $\sigma$ | 2.92 | 1.89 | 3.75 | 4.16 | 3.76 | 2.34 | 0.00 | 0.25 | 0.40 | 0.05 | 2.47 | 0.51 | 4.07 |

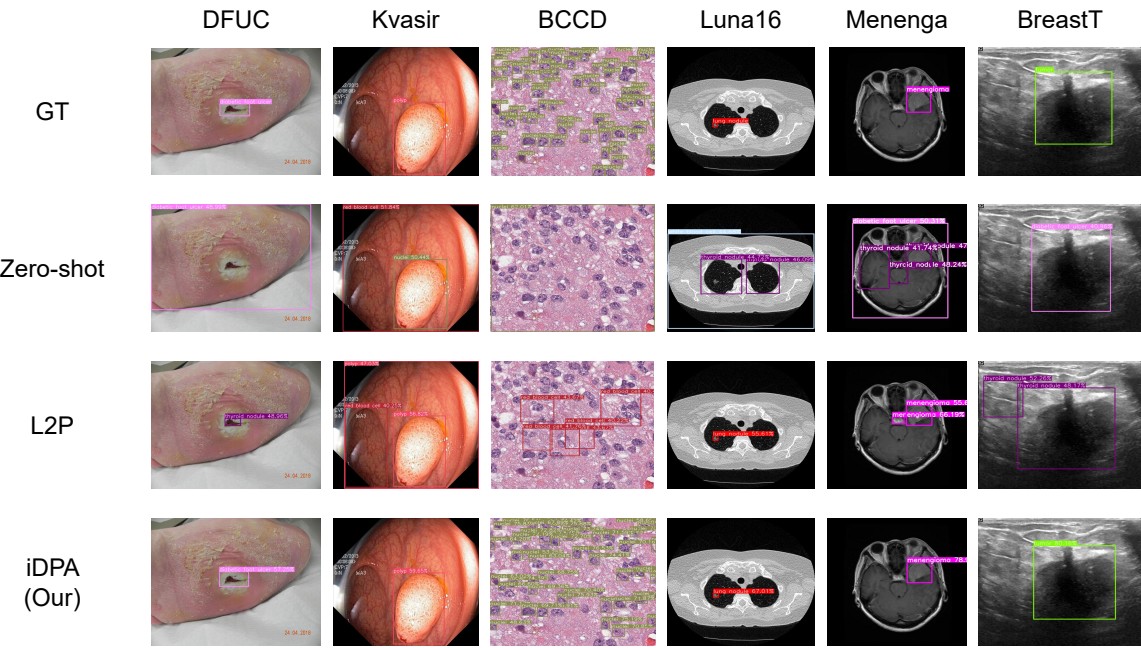

**Text Prompt:** "diabetic foot ulcer. polyp. optic nerve. platelets. red blood cell. white blood cell. nuclei. breast tumor. tuberculosis. kidney tumor. lung nodule. hippocampus. menengioma. tumor. thyroid nodule. "

Figure 7: Visualization results of iDPA compared with L2P and Zero-shot at the end of training with random seed 0 on ODinM-13.

