# OpenReview forum: "iDPA: Instance Decoupled Prompt Attention for Incremental Medical Object Detection"
_ICML.cc/2025/Conference — ICML 2025 poster_

### Official Review · Reviewer_VMtc · 2025-03-10

**Overall Recommendation:** 3

**Summary:**

This paper proposes a novel framework, instance Decoupled Prompt Attention (iDPA), for incremental object detection in medical images. This task is challenging due to the strong coupling between foreground-background features and the large domain gap between natural and medical images. This work proposes instance-level prompt generation (IPG) and decoupled prompt attention (DPA) to more effectively leverage the medical object information and prompt knowledge. Extensive comparison experiments on full dataset and few-shot settings demonstrate the superiority of the proposed method over the existing incremental object detectors.

## update after rebuttal
The authors added the experiments to address most of my concerns. I also read the discussions between the other reviewers and the authors. I appreciate the efforts that authors compare the proposed method with SAM2.1-L and discuss the model efficiency. Overall, I would like to keep my positive score.

**Claims And Evidence:**

Yes, the claims in this paper are supported by its experiments.

**Essential References Not Discussed:**

No

**Experimental Designs Or Analyses:**

I have checked the validity of the experiment designs and found them reasonable.

**Methods And Evaluation Criteria:**

This paper proposed a new method for incremental medical object detection. A large dataset for this task is built upon 13 existing datasets to evaluate its method.

**Other Comments Or Suggestions:**

No

**Other Strengths And Weaknesses:**

Strengths:

1)	This paper is written clearly and easy to follow.

2)	This method designs the knowledge decoupling from the instance level and generates more precise concept prompts than the mixed knowledge used in classification tasks.

3)	A large-scale medical detection dataset, including 13 tasks, for incremental learning is proposed to promote the development of this research field.

Weakness:

1)	Different medical image modalities have various visual characteristics, and different diseases appear in various body regions. The image background information is important for the medical detection task, especially when the model weights are transferred from the natural image domain. Why does IPG only use the features around and within the bounding boxes? Will this design decrease model robustness to the slight domain shift?

2)	An ablation study needs to be conducted on the scaling factor in Eq. (6).

3)	In Eq. (8), what do the symbols of W_k and W_v represent?

4)	In IPG, why is the CCPKI for the i-th task only initialized from the i-1-th task instead of all tasks before the i-th one?

5)	It remains unclear in DPA whether the performance improvement is primarily attributed to Eq. (10), removing the learning of [¯p_t], or Eq. (13), learning new parameters $\lambda$. We recommend conducting more detailed ablation studies on the DPA modules.

6)	In Table 3, why does the performance of the naïve baseline already outperform most of the methods in Table 1? Do the authors conduct grid search for all comparison methods to choose their best learning rates?

**Questions For Authors:**

Please solve the questions in the weakness.

**Relation To Broader Scientific Literature:**

This work is meaningful for building stronger tools with continual learning abilities for medical image analysis.

**Theoretical Claims:**

I have checked the formulas in this paper and found no errors.

---

> ### Author Rebuttal · Authors · 2025-04-01
>
> We appreciate your detailed feedback and suggestions for improvement. We treasure the opportunity to address your concerns and improve our work.
>
> ##  Weakness 1: Domain Shift Robustness with Bounding Box-only Features
> Thank you for this important observation. The paper focuses emphasizes instance-level knowledge decoupling to specifically focus on discriminative foreground regions (e.g., lesions or organs), thereby minimizing interference from irrelevant background information. The inherent cross-modal alignment capabilities of VLOD models such as GLIP also help in capturing contextual relationships, which further reduces dependency on background features. Empirical results in Table 1 demonstrate that our proposed iDPA consistently outperforms global prompt-based methods (e.g., L2P, DualPrompt) in full-data settings, indicating that localized feature extraction improves detection accuracy without compromising robustness to domain shifts. Moreover, Figure 5(a) empirically validates that instance-level knowledge surpasses image-level knowledge in our scenario. Nevertheless, we agree with the reviewer that selectively incorporating relevant background context could further enhance the model’s robustness. Thus, in future work, we plan to explore hybrid attention mechanisms that effectively integrate valuable background information during the feature-to-prompt knowledge transfer process.
>
> ## Weakness 2: Lack of Ablation on Scaling Factor α in Eq. (6)
> We would like to thank the reviewer for the suggestion. We have conducted new experiments with respect to $\gamma$, as shown in the table below:
> | Scaling factor | FAP ↑  | CAP ↑  | FFP ↓ |
> |----------------|--------|--------|-------|
> | 1.00           | 50.07  | 54.01  | 2.57  |
> | 1.30           | 50.28  | 54.10  | 2.48  |
> | 1.50           | 49.99  | 53.40  | 2.77  |
>
> ## Weakness 3: Ambiguity in W_k and W_v in Eq. (8)
> W_k and W_v are **linear projection layers** mapping instance features (v_c) into query/key/value spaces for cross-attention. They enable adaptive feature alignment between instance representations and task-specific prompts.  These will be clearly defined in the revised manuscript.
>
> ## Weakness 4: Limited Initialization Scope in CCPKI
>
> Incremental initialization (from task $i-1$) balances the stability-plasticity trade-off: recent tasks are given higher priority to mitigate forgetting, while older tasks are retained through the prompt pool.
>
> Compared to using "all tasks before the $i$-th one," this approach reduces the number of additional model parameters. By using the $i-1$-th task for initialization, model training parameters remain consistent, requiring only $1 \times \theta^{CCPKI}_{i-1}$.
>
> In contrast, using all tasks prior to the $i$-th task would require a total of $i \times \theta^{CCPKI}_{i-1}$ additional parameters.
>
> ## Weakness 5: Unclear Contribution of DPA Components
> To clarify whether DPA's gains stem from removing the $[\overline{p_t}]$ learning (Eq. 10) or introducing $\lambda$ (Eq. 13), we present this in Fig. 5(b). Compared to the naive method, removing $[\overline{p_t}]$ (i.e., when the scale type is 1.0) results in a +2.79\% FAP, while our final approach, which introduces $\lambda$ (dim), leads to a +5.29\% FAP.
>
> ## Weakness 6: Unfair Comparison of Naïve Baseline in Table 3
>
> We appreciate the reviewer’s concern. However, due to hyperparameter differences, the Naïve baseline in Table 3 operates on $\Phi_{f}$, which requires fewer parameters compared to $(\Phi_{v} + \Phi_{t})$, resulting in lower memory usage and reduced training time. In contrast, the methods in Table 1 operate on $(\Phi_{v} + \Phi_{t})$. To further clarify, we have added the Naïve method for $(\Phi_{v} + \Phi_{t})$, which achieves 41.72 FAP, 47.30 CAP, and 8.57 FFP. By introducing the iDPA method on top of the Naïve approach for $(\Phi_{v} + \Phi_{t})$, we significantly improve these metrics, with FAP increasing by 6.69, CAP by 6.26, and FFP decreasing by 4.19. These improvements, compared to the methods in Table 1, highlight that the gains are due to the effectiveness of our approach, rather than hyperparameter tuning discrepancies.
>
> | $(\Phi_{v} + \Phi_{t})$ | FAP ↑  | CAP ↑  | FFP ↓ |
> |------------------------|--------|--------|-------|
> | Naïve | 41.72  | 47.30  | 8.57  |
> | iDPA (ours)            | 48.41  | 53.56  | 4.38  |
> | $\Delta$               | 6.69   | 6.26   | 4.19  |

---

> > ### Comment · Reviewer_VMtc · 2025-04-04
> >
> > Thank you for the detailed reply. The new results and discussion of the hyperparameter differences in reply 6 should be added to the final version of this work.

---

> > > ### Author Response · Authors · 2025-04-07
> > >
> > > Thank you for the suggestion. We will incorporate the new results and the discussion on hyperparameter differences from reply 6 into the final manuscript for clarity and completeness.

---

### Official Review · Reviewer_HWoG · 2025-03-10

**Overall Recommendation:** 3

**Summary:**

This paper aims to tackle the challenge of incremental medical object detection, which adapts to emerging medical concepts and retains prior knowledge. The authors claim that existing works are only designed for classification and fail to capture fine-grained information for detection tasks, which are mainly limited to the (a) coupling between foreground-background information and (b) coupled attention between prompts and image-text tokens. To tackle these challenges, this paper proposes an iDPA framework consisting of an instance-level prompt generation to enhance dense predictions. Besides, It introduces a decoupled prompt attention to enhance the knowledge transfer of prompts. Experiments demonstrate that iDPA achieves superior performance in both full-data and few-shot settings while being efficient regarding trainable parameters and memory usage.

**Claims And Evidence:**

No.
- The mentioned conceptual gap between medical and natural domains is unclear. There is no experimental or theoretical to justify this gap. Besides, I cannot find any convincing designs tailored for the medical imaging, and believe that this method can also work for generic domains.
- The value of incremental medical object detection is unclear. Many medical foundational models, like medical SAM, can achieve superior zero-shot performance on new datasets. What is the value of designing a handcrafted incremental setting instead of scaling up datasets for the model training?

**Essential References Not Discussed:**

There is no sufficient discussion about the continual learning in dense prediction [1,2].

[1] Eclipse: Efficient continual learning in panoptic segmentation with visual prompt tuning CVPR 24
[2] A survey on continual semantic segmentation: Theory, challenge, method, and application TPAMI 24

**Experimental Designs Or Analyses:**

The experimental design is sound by considering the full-data and few-shot settings, with comparisons against state-of-the-art methods and ablation studies for key components. However, there are many issues:

- There are no comparisons with state-of-the-art methods regarding computational efficiency and memory usage.
- There is also no discussion about the experiments in Sec.5.4, leading to the limited experimental insight.
- There lacks a comparison with some latest continual learning works focusing on dense prediction [1,2]

[1] Eclipse: Efficient continual learning in panoptic segmentation with visual prompt tuning CVPR 24
[2] A survey on continual semantic segmentation: Theory, challenge, method, and application TPAMI 24

**Methods And Evaluation Criteria:**

Even though the proposed method lacks convening evidence tailored for medical domains, the technical parts of decoupling the prompts and enabling fine-grained representation are reasonable and technically sound.

**Other Comments Or Suggestions:**

Why the sentence in Sec 4.1. is in blue color?

**Other Strengths And Weaknesses:**

Strength
- The proposed methods exploring box-level prompts make sense and interesting
- The paper is easy to follow


Weakness
- Lack some comparison with the continual learning works focusing on the dense predictions.
- Lack the convening evidence and motivation to tackle the issue for medical domains instead of the generic domain
- The model efficiency may be influenced since the method requires GLIP to generate local-level RoIs. It is necessary to compare this with other methods of efficiency.

**Questions For Authors:**

- Since GLIP is trained in the generic domain, how can the quality of generated RoIs be ensured?
- In Tab.1, using all data to train a unified model gives the best performance. So, how to justify the clinical value of the continual learning setting in the medical domain?
- How about the model efficiency compared with state-of-the-art works, such as model parameters and inference speed?
- The mentioned conceptual gap between medical and natural domains is unclear. How does the proposed method address this gap? The natural domain seems more challenging since it consists of more diverse object classes, appearances, and scales. Besides, the authors used GLIP pre-trained in the natural domain, which has a gap with the medical domain if the claimed gap exists.
- How about the comparison with the dense prediction methods in continual learning?

**Relation To Broader Scientific Literature:**

The proposed method may provide some insights into generic object detection, continual learning, and some scenarios requiring fine-grained visual evidence.

**Theoretical Claims:**

The paper does not have proofs or theoretical claims. I have checked the equations in the methodology and did not find significant mistakes.

---

> ### Author Rebuttal · Authors · 2025-04-01
>
> **Question 1: Conceptual Gap Between Medical and Natural Domains**
>
> We appreciate the reviewers' feedback and would like to clarify the differences between the medical and natural domains, as well as the limitations of existing methods in medical object detection. Previous studies, such as those by Qin et al. (2022), Ma et al. (2024), and Zhu et al. (2024), highlight key challenges in the medical domain, including data scarcity, modal diversity (e.g., CT, MRI, X-rays), and annotation complexity, as medical annotations require expert input. Existing methods face issues like foreground-background coupling, where background areas may confuse classifiers, and prompt-label attention coupling, which can dilute prompt information and reduce sensitivity to subtle features, such as small lesions. The iDPA framework overcomes these challenges through instance-level prompt generation and decoupling attention, significantly improving medical detection. While designed for medical applications, our method shows strong generalization capabilities, suggesting it can also perform effectively in natural domains due to the greater complexity of the medical domain. We hope this clarifies the necessity and relevance of our method in the medical field, and we appreciate the reviewer’s suggestions, which have helped refine the paper.
>
> **Question 2: Value of Incremental Medical Object Detection**
>
> Incremental medical object detection is crucial for addressing real-world challenges in clinical deployment, offering significant advantages over zero-shot models like Medical SAM. It enables adaptation to new tasks without full retraining, thus overcoming regulatory constraints (e.g., HIPAA, GDPR), and supports the dynamic evolution of medical knowledge, allowing models to adjust to new diseases and imaging technologies without losing prior learning. Incremental learning effectively handles the diversity of medical image modalities (CT, MRI) and annotation inconsistencies, ensuring generalization across different types while reducing the computational and storage demands of retraining large models. While models like Medical SAM excel in zero-shot scenarios, they struggle with rare, out-of-distribution concepts and multitask compatibility. Incremental learning methods like iDPA are better suited for these challenges, particularly in detection tasks. Our method complements models like MedSAM by generating bounding boxes as prompts for precise predictions and enabling continuous learning in out-of-distribution domains. In summary, incremental medical object detection addresses regulatory, data, modality, and learning challenges in dynamic medical settings.
> - Qin, Ziyuan, et al. "Medical image understanding with pretrained vision language models: A comprehensive study." arXiv preprint arXiv:2209.15517 (2022).
> - Ma, Jun, et al. "Segment anything in medical images." Nature Communications 15.1 (2024): 654.
> - Zhu, Jiayuan, et al. "Medical sam 2: Segment medical images as video via segment anything model 2." arXiv preprint arXiv:2408.00874 (2024).
>
> **Question 3: Comparison with Dense Prediction Continual Learning Works**
>
> The paper focuses on incremental object detection (bounding box regression and classification) in medical images, with an emphasis on dense prediction approaches like DenseBox, which predict relative object positions. This differs from tasks like panoptic or open-vocabulary segmentation. iDPA contrasts with methods such as Eclipse (CVPR 2024) and PanopticCLIP (TPAMI 2024): Eclipse targets panoptic segmentation with dense pixel masks, while iDPA detects discrete objects through bounding boxes. Both methods use prompts, but iDPA’s instance-level prompt generation isolates fine-grained features, which could inspire dense prediction methods, though direct comparison is challenging due to task differences. PanopticCLIP focuses on zero-shot open-vocabulary segmentation, while iDPA handles incremental class learning. Technically, Eclipse uses spatial-semantic cues for segmentation, whereas iDPA decouples instance-level knowledge from background clutter. Architecturally, Eclipse modifies segmentation heads like Mask2Former, while iDPA extends VLOD models like GLIP. Future work includes using iDPA’s decoupled prompt attention to enhance dense prediction and combining it with panoptic prompts for joint detection-segmentation continual learning. We also plan to evaluate iDPA on dense prediction datasets (e.g., Retina, ISIC) to assess its generalizability.
>
> **Question 4: Others**
> 1. We will add an efficiency comparison of the method in Table 1 in the revised manuscript, where the high efficiency of our method can already be observed in the current table.
>
> 2. During training, the ground truth (gt) boxes are used to generate instance features, and during testing, only the prompts saved in the prompt pool are used.
>
> 3. The sentence in Section 4.1 is in blue color to indicate emphasis.

---

> > ### Comment · Reviewer_HWoG · 2025-04-05
> >
> > Thanks for the rebuttal. However, most of my concerns have not been addressed.
> >
> > (1) The authors claimed the large gap between the natural and medical domains in Q1. However, they use GLIP trained in the natural domain without any medical knowledge and find large improvements, which contradicts the author's claim. There are no convincing experiments to solve this concern.
> >
> > (2) The authors do not give any response about the comparsion in model efficiency with other methods, which is a critical aspect in clinical application.
> >
> > (3) There is no convincing explanation why the authors do not compare with dense prediction methods. Only comparing with classification methods makes the comparison obviously unconvincing. Besides, object detection is a sub-task in instance segmentation and panoptic segmentation. I have no idea why the comparison cannot be done.
> >
> > Besides, after reading other rebuttals and reviews, I find lots of other concerns are not well addressed obviously. Hence, based on the unconvincing rebuttal, I will decrease my score and recommend rejecting this paper.

---

> > > ### Author Response · Authors · 2025-04-07
> > >
> > > We sincerely thank the reviewer for the thorough follow-up and continued engagement with our work. We regret that our previous rebuttal did not fully address your concerns and appreciate the opportunity to provide further clarification. Below, we respectfully elaborate on the key points raised:
> > >
> > > **(1) On the claimed gap between natural and medical domains vs. use of GLIP:**
> > >
> > > We understand the reviewer’s concern and would like to clarify the nuance of our claim. Our argument is not that GLIP is already optimal for medical domains, but rather that when guided by carefully designed prompts and decoupled attention mechanisms, its general capabilities can be repurposed to benefit medical detection，e.g., almost 0% AP in original GLIP vs > 50% AP in GLIP+medical engineered prompts. This demonstrates the potential to bridge the domain gap, not a contradiction of its existence.
> > >
> > > **(2) On the model efficiency comparison:**
> > >
> > > In our previous rebuttal, we stated that a detailed efficiency comparison would be incorporated into the revised manuscript in Table 1. Furthermore, our existing results in Tables 3 and 4 already showcase iDPA’s high efficiency, particularly in terms of trainable parameters and memory usage. To reiterate, our experiments consistently demonstrate that iDPA achieves a superior balance of accuracy and efficiency compared to baseline methods. For the reviewer’s convenience and to ensure completeness, we provide the detailed efficiency comparison below:
> > >
> > > | Methods          | #Params↓ | #Memory↓ | #Time↓  | Inference Speed↑ | FAP↑   |
> > > |------------------|----------|----------|---------|------------------|--------|
> > > | Joint (Upper)    | 231.76M  | 13129M   | 9h55min | 6.18             | 54.67  |
> > > | Sequential       | 231.76M  | 13129M   | 9h55min | 6.18             | 4.4    |
> > > | WiSE-FT          | 231.76M  | 13129M   | 9h55min | 6.18             | 10.72  |
> > > | ER               | 231.76M  | 13129M   | 11h15min| 6.18             | 39.91  |
> > > | ZiRa             | 10.23M   | 8377M    | 6h25min | 6.11             | 3.66   |
> > > | L2P              | 6.97M    | 10288M   | 7h50min | 5.08             | 39.88  |
> > > | DualPrompt       | 4.83M    | 9417M    | 7h36min | 5.25             | 28.89  |
> > > | S-Prompt         | 2.73M    | 5366M    | 8h24min | 5.13             | 41.02  |
> > > | CODA-Prompt      | 10.97M   | 9803M    | 9h03min | 5.26             | 42.08  |
> > > | DIKI             | 8.76M    | 9754M    | 7h49min | 5.16             | 42.51  |
> > > | NoRGa            | 8.76M    | 9963M    | 8h07min | 5.17             | 44.84  |
> > > | Ours             | 3.34M    | 6590M    | 5h46min | 5.93             | 50.28  |
> > >
> > >
> > > **(3) On comparison with dense prediction methods:**
> > >
> > > * 1. Task Distinctions: Object detection, instance segmentation, and panoptic segmentation are distinct yet complementary visual tasks. Object detection (e.g., DETR, DINO) focuses on localizing objects with bounding boxes and identifying their categories, while instance segmentation and panoptic segmentation go further by generating pixel-level masks (e.g., Mask2Former) and distinguishing between objects. Detection models like YOLO independently produce bounding boxes without requiring a segmentation module, whereas segmentation models rely on additional components (e.g., Mask Head) to generate masks, optimizing for metrics like mIoU, which differ from detection metrics such as mAP. Although these tasks overlap to some extent, they fundamentally differ in their objectives, model architectures, and outputs. Forcing object detection to be categorized as a subset of segmentation risks obscuring its core characteristics.
> > > * 2. Task Complexity vs. Generality: The hierarchy of task complexity, **panoptic segmentation > instance segmentation > object detection > image classification**, reflects increasing demands on model architecture and specialization. However, this specialization often sacrifices generality, there is no free lunch. For example, dense prediction continual learning methods like ECLIPSE are more specialized than object detection counterparts like iDPA. ECLIPSE classifies each category prompt as object or non-object, leverages prior knowledge that old categories may serve as background for new ones, and uses mutual information to generate a refined no-object logit, helping to mitigate error propagation and semantic drift. In contrast, iDPA, built on GLIP, directly computes classification scores via dot products between visual and textual features, without object/non-object prompt separation.
> > > * 3. Moreover, we are not only comparing with classification-based methods, but also with vision-language model-based continual object detection methods of the same type, such as ZiRa [NeurIPS 2024].
> > >
> > > Finally, we sincerely appreciate the reviewer’s re-evaluation of our work and the increased score. We are grateful for the time and effort dedicated to reviewing our manuscript and for your thoughtful consideration and constructive feedback.

---

### Official Review · Reviewer_Sm8X · 2025-03-13

**Overall Recommendation:** 3

**Summary:**

This paper proposes a novel incremental medical object detection framework called iDPA, which is composed of an instance-level prompt generation (IPG) and a decoupled prompt attention (DPA) module. Comparing existing methods, the instance-level prompt generation provides learnable prompts with fine-grained task-specific knowledge, and the DPA module helps simplify the prompt attention mechanism and mitigates the forgetting issue during the task transfer. The proposed iDPA methods show a consistent improvement in incremental learning settings of 13 different tasks, outperforming existing SoTA in both full-data and few-shot settings.

**Claims And Evidence:**

Yes, most of the paper's claims are properly supported by the experimental results. The ablation experiments on page 8 help illustrate each module's contribution and provide an empirical reason for the model design. The experiment also validates the DPA module's memory and speed improvement.

**Essential References Not Discussed:**

The reviewer didn't find such a missing reference in the paper.

**Experimental Designs Or Analyses:**

Yes, the evaluation in the paper is generally convincing and thorough.
1. The improvement of the proposed method is non-trivial compared with the SoTA baseline.
2. The paper has also provided detailed implementation details of the evaluation and evaluated the proposed method under different settings.
3. The ablation experiment also helps understand the effectiveness of each method.

Yet, the reviewer does notice a small issue in the abstract. The performance improvement reported in the paper is "5.44%, 4.83%, 12.88%, and 4.59%" for each setting, but the results in the abstract on the openreview is "5.44%, 4.83%, 15.39%, and 4.59%", where the third value is different. But I assume this is just a typo.

**Methods And Evaluation Criteria:**

Yes, the proposed IPG and DPA methods are intuitively reasonable and were validated empirically through experiment. Using instance-level prompts can naturally provide fine-grained task-specific information while the DPA module is also proven to be efficient and effective. Based on the evaluation in Tables 1 and 2, it is clear that the proposed method outperforms existing SoTA with a non-trivial gap, and it is also meaningful for the future development of the domain.

## Weakness:

1. The reviewer's major concern here is the significance of the incremental learning setting, especially for the full data evaluation. Since the full data is available, will the performance of the continue learning-based method still be better than the regular task-specific model? Additionally, how about the performance of all-in-one style medical SAM [a] foundation models? But this will not harm the novelty and soundness of the paper.
2. Another specific concern for the iDPA is the design of instance-level prompt generation. According to section 4.2, the instance-level prompt is generated via the instance-level image feature, which needs training data from the bbox to extract these instance-level features. This is fine for the full data setting, but the reviewer worries that this will be a problem in the few-shot settings, where the training data may not be able to provide a set of high-quality instances. There is no evaluation on the stability of the proposed model on few-shot settings as well.
3. The last minor concern is the residual connection in each component. Most of the proposed module is connected to the main framework via a residual connection, which can potentially weaken the contribution. Yet, the ablation experimental results show the effectiveness of each component, so it is just a minor issue.

[a] Zhu, Jiayuan, et al. "Medical sam 2: Segment medical images as video via segment anything model 2." arXiv preprint arXiv:2408.00874 (2024).

**Other Comments Or Suggestions:**

N/A

**Other Strengths And Weaknesses:**

The reviewer does find two additional weaknesses for the proposed method.

1. The overall complexity of the method. The proposed iDPA is composed of two different modules, each with a set of complex designs and hyper-parameters that can be tuned in application. The ablation experiment helps discuss the effectiveness of each module. The complexity is still concerning.
2. A major issue, as for the reviewer, is the writing of the paper. There are multiple equations and notations in the paper, but they are not all properly defined. Some of variable even uses repeated notations and make it confusing to follow the paper. Additionally, Figure 2 is also kind of messy. This really hinders the readers from understanding the paper. A few examples are listed below:
    - Both the centriod of the features and the number of instant-level representations use the letter K as their notation.
    - The part of section 4.2 between line 200 to 219 is particularly unclear.
    - The weight of each linear projection layer seems to be represented using W in the equation but never defined.
    - As for the CCPKI module, it only discussed the creation of prompt $p_i$, where $i$ is the i-th prompt. However, there is no discussion about the difference between $p_v$ and $p_t$, as shown in Figure 2. It is unclear if $p_v$ and $p_t$ are generated using the same mechanism or not.
    - The activation function $\sigma$ in equation (8) is not defined, though the reviewer guessed it might be the sigmoid activation.
    - The notation in equation (9) is also confusing. The double vertical line usually serves for the norm between two vectors, but it seems to be a bracket in equation (9), which is very confusing.
    - Equation (10-13) uses different colors for some of the components in the equation; it is actually not that clear what the meaning of these colors is.

**Questions For Authors:**

1. The reviewer is not sure what Table 5 is evaluating. According to the method section and Figure 2, the DPA module is only inserted once in each encoder. What is the meaning of the different X-Attn number here?
2. According to Figure 2, the DPA module is applied for all three encoders, but why is the row with only the fusion encoder colored in red in Table 4?
3. In the DPA module, is the attention computed separately for each pair of inputs? Will this cause additional memory access during computation? The regular PA seems to need just one large matrix multiplication, while DPA needs three.

**Relation To Broader Scientific Literature:**

The proposed method mainly focuses on incremental learning for medical object detection settings. It is developed based on the existing framework of GLIP and swin-transformer. The design of IPG and DPA is also a modification of the existing learnable prompt pool for each task and the prompt attention mechanism in the previous works.

In this paper, the proposed method mainly focuses on optimizing the prompt generation and attention mechanism, providing a more specific and fine-grained prompt for each task. Also, improves efficiency via using decoupled prompt attention.

**Theoretical Claims:**

N/A, the paper didn't propose a new theoretical claim. The proposed method is evaluated through experiments empirically.

---

> ### Author Rebuttal · Authors · 2025-04-01
>
> **Question 1: Significance of Incremental Learning in Full-Data Settings**
>
> Incremental learning methods like iDPA offer significant advantages in medical settings where models must be deployed incrementally due to regulatory, ethical, or resource constraints. Retraining large models on full datasets is often too costly, but iDPA only updates prompt vectors (1.4% of trainable parameters), which reduces training time. While Medical SAM performs well in zero-shot segmentation, it struggles with long-tail classes and domain shifts. On the other hand, iDPA enables incremental class learning and preserves prior knowledge, effectively handling challenges such as adding new classes over time. It also supports task-specific adaptation, such as joint detection and classification, which SAM is not designed for.
>
> **Question 2: Stability of Instance-Level Prompt Generation in Few-Shot Settings**
>
> The IPG module ensures stability in few-shot settings by aggregating features from all instances within a class using cross-attention, reducing reliance on noisy instances. It combines local features with global ones, preventing overfitting to sparse annotations. Empirical results show that iDPA outperforms other methods in 1-shot settings and provides a 3.66% FAP improvement compared to naive prompts. IPG also enhances generalization by using a scaling factor (γ) to expand regions of interest (RoIs), capturing contextual information beyond tight bounding boxes. By leveraging pretrained GLIP model knowledge, IPG prevents overfitting even with few samples. To further improve stability, semi-supervised initialization, hard negative mining, and hyperparameter tuning are suggested.
>
> **Question 3: Model Complexity and Writing Clarity**
>
> Thank you for the reviewer’s valuable feedback. We appreciate the concerns regarding model complexity and writing clarity. iDPA’s modular design was intentionally chosen for flexibility, with IPG handling instance-level knowledge extraction and DPA optimizing attention. To manage complexity, we have designed hyperparameters like K and λ to be learned adaptively and are exploring simplifications such as hyperparameter-free designs. We also recognize the importance of clear and consistent notation and will standardize symbols and provide clearer explanations for equations, ensuring the content remains both accessible and rigorous.
>
> **Question 4: Technical Details of DPA Module**
> 1. **Table 5 Interpretation**:
>    - The "X-Attn Number" represents the number of cross-attention layers in the fusion encoder where DPA is applied. For example, "6 (all)" indicates that DPA is enabled in all 6 layers.
>
> 2. **Figure 2 Redesign**:
>    - It should be clarified that DPA is applied to all three encoders (visual, text, and fusion), with emphasis on its role in the fusion stage. The visual and text encoders are optional, as shown in the ablation experiments in Table 4. When all three are used together, continual learning performs best. However, incorporating DPA into the fusion encoder provides a good balance of performance, additional parameter load, memory usage, and training time. Therefore, in subsequent experiments, we default to using the fusion encoder.
>
> 3. **Computational Efficiency**:
>    - **Memory Access**: DPA requires three parallel attention computations (vision→text, text→vision, and original PA). However, this is mitigated by:
>      - **Reduced Feature Length**: The prompt length (l=10) is much shorter than image/text tokens (L=10,000+), which minimizes memory overhead.
>      - **Computation Merge**: During testing, we merge the three parallel attention computations, reducing the overall computational cost compared to the original PA. This is further demonstrated in Reviewer rTU6 under the section "Mathematical Justification for DPA Superiority in Computation Cost."

---

> > ### Comment · Reviewer_Sm8X · 2025-04-03
> >
> > I appreciate the effort made by authors during the rebuttal period, and also glad to see the new discussions, like the motivation of Incremental learning and clarification about the DPA module.
> >
> > However, my concern is not fully addressed yet.
> >
> > 1. There is still no comparison with one-for-all style baselines like MedSAM. While the explanation is promising, I would still like to see some quantitative results to support this claim. I also notice a similar concern by the reviewer HWoG.
> > 2. As for the stability in the few-shot settings, it would be better if providing some sort of variance measure of the performance under multiple few-shot runs with different contexts (different training data), which shouldn't take too much effort. I acknowledge the explanation provided by the authors, but additional results are expected here to better support the claim.
> > 3. Although the author claims they will aim to develop parameter-free methods in the future, the complexity issue remains. Still, the proof provided in the reply to reviewer rTU6 is helpful here.
> >
> > Overall, the rebuttal provides some intuitive explanation and discussion of my concern, but the lack of quantitative results makes it less persuasive. Thus, I choose to maintain my score of 3 here.

---

> > > ### Author Response · Authors · 2025-04-08
> > >
> > > We sincerely thank the reviewer for the detailed feedback. We regret that our previous rebuttal did not fully address your concerns and appreciate the opportunity to clarify further.
> > >
> > > **1. Response to One-for-All Style Comparisons**
> > >
> > > We appreciate your suggestion regarding one-for-all style comparisons. We have been working to reproduce MedSAM2 and are grateful for the open sharing of its code and weights. However, the 2D pre-trained weights are still under development (as mentioned in [this issue](https://github.com/SuperMedIntel/Medical-SAM2/issues/8#issuecomment-2291242296)), and we encountered difficulties adapting the 3D weights to our task (similar to the issue discussed in [this link](https://github.com/SuperMedIntel/Medical-SAM2/issues/9)). Although we attempted to re-train MedSAM2 on 2D data, the process was time-consuming and could not be completed within the rebuttal period. As a result, we decided to use SAM2.1-L model weights, which retain MedSAM2's core modules (such as the self-sorting memory bank and interval click) for comparison. The results are shown in the table below:
> > >
> > > | Method | DFUC | Kvasir | OpticN | BCCD | CPM-17 | BreastC | TBX11K | KidneyT | Luna16 | ADNI | Meneng | BreastT | TN3k | FAP ↑ |
> > > | --- | --- | --- | --- | --- | --- | --- | --- | --- | --- | --- | --- | --- | --- | --- |
> > > | SAM2.1 | 0.06 | 1.71 | 0.00 | 8.83 | 18.00 | 0.15 | 0.00 | 0.67 | 0.23 | 0.00 | 12.54 | 0.08 | 0.22 | 3.27 |
> > > | MedSAM2* | 4.15 | 5.34 | 4.53 | 13.83 | 22.44 | 1.54 | 0.00 | 1.56 | 4.32 | 0.04 | 18.56 | 2.96 | 3.45 | 5.02 |
> > > | Ours | 47.09 | 73.76 | 66.85 | 60.29 | 36.54 | 50.98 | 32.69 | 64.98 | 31.15 | 44.42 | 57.20 | 34.65 | 53.03 | 50.28 |
> > >
> > > Here, SAM2.1 refers to the auto-segmentation results, and MedSAM2* refers to the results with the SAM2.1-L model.
> > >
> > > **2. Variance in Few-Shot Settings**
> > >
> > > We appreciate your suggestion to include variance in the few-shot settings. We have now added variance results in the table below to demonstrate the stability of our method across multiple runs:
> > >
> > > | Shot | DFUC | Kvasir | OpticN | BCCD | CPM-17 | BreastC | TBX11K | KidneyT | Luna16 | ADNI | Meneng | BreastT | TN3k |
> > > | --- | --- | --- | --- | --- | --- | --- | --- | --- | --- | --- | --- | --- | --- |
> > > | 1 | 6.66 | 43.04 | 14.62 | 20.55 | 31.13 | 5.33 | 2.15 | 7.38 | 0.30 | 0.39 | 17.34 | 6.17 | 3.45 |
> > > |$\Delta$ | 2.55 | 0.95 | 4.89 | 3.86 | 1.71 | 0.93 | 0.00 | 1.55 | 0.14 | 0.03 | 2.18 | 2.46 | 1.31 |
> > > | 5 | 21.37 | 50.20 | 29.20 | 39.11 | 38.33 | 19.65 | 6.03 | 27.06 | 6.23 | 3.15 | 39.42 | 15.34 | 14.02 |
> > > | $\Delta$ | 1.62 | 0.86 | 3.70 | 3.24 | 2.63 | 1.24 | 0.01 | 0.02 | 1.68 | 1.26 | 1.34 | 1.58 | 3.80 |
> > > | 10 | 34.79 | 59.03 | 52.64 | 58.12 | 39.33 | 37.35 | 14.78 | 52.77 | 22.70 | 24.55 | 56.32 | 10.99 | 39.10 |
> > > |$\Delta$ | 2.92 | 1.89 | 3.75 | 4.16 | 3.76 | 2.34 | 0.00 | 0.25 | 0.40 | 0.05 | 2.47 | 0.51 | 4.07 |
> > >
> > > **3. Complexity**
> > >
> > > Regarding the complexity, we acknowledge that it remains a challenge. While we aim to develop parameter-free methods in the future, the current approach has been validated through ablation studies, showing the necessity of each module. In future work, we plan to optimize the method further by reducing hyperparameter tuning, such as through automatic instance selection with the self-sorting memory bank in the IPG module, ensuring high-quality instance efficiency. Besides, in our latest discussion with Reviewer rTU6, we have added a FLOPs comparison metric to address the concerns raised. We hope this can further clarify and resolve any doubts regarding the computational efficiency of our method.
> > >
> > >
> > >
> > > We sincerely thank the reviewer for the valuable feedback, which has significantly improved the quality of our paper. Your comments were crucial in refining our approach, and we appreciate the time and effort you’ve dedicated to reviewing our work. We hope the clarifications and additional results provided address your concerns and offer a clearer understanding of our contributions and computational advantages. Given these updates, we kindly ask if you could reconsider the score.
> > >
> > > Thank you again for your thoughtful consideration.

---

### Official Review · Reviewer_rTU6 · 2025-03-14

**Overall Recommendation:** 3

**Summary:**

This paper proposes iDPA (Instance Decoupled Prompt Attention), a novel framework for Incremental Medical Object Detection (IMOD). The primary motivation is that existing prompt-based continual learning methods, while effective for classification tasks, struggle with object detection due to the need for fine-grained instance-level reasoning.

To this end, the authors introduce:
- Instance-level Prompt Generation (IPG): A mechanism to decouple fine-grained instance knowledge from images and generate prompts that better focus on dense medical object detection.
- Decoupled Prompt Attention (DPA): A modification of standard prompt attention mechanisms, separating prompt-token interactions to enhance knowledge transfer, reduce memory overhead, and mitigate catastrophic forgetting.

The authors construct ODinM-13, a benchmark of 13 cross-modal, multi-organ, multi-category medical datasets, and demonstrate that iDPA outperforms state-of-the-art (SOTA) methods in full data and few-shot settings (1-shot, 10-shot, 50-shot).

**Claims And Evidence:**

**Lacking evidence in generality beyond ODinM-13**: The model is only evaluated on ODinM-13; additional benchmark testing (e.g., public MOD datasets) would strengthen the claim of generalizability.

**Mathematical justification for DPA superiority**: While the scaling factor λ(ft) is well-motivated, additional formal proofs explaining why DPA improves prompt learning over standard Prompt Attention (PA) would be beneficial.

**Essential References Not Discussed:**

Not familiar with the latest works.

**Experimental Designs Or Analyses:**

Lack of proof for DPA’s efficiency: a computational complexity comparison between DPA and standard prompt attention is expected.

**Methods And Evaluation Criteria:**

Strengths:
- ODinM-13 provides a **diverse, realistic benchmark** for incremental medical object detection.
- **Comprehensive baselines**: The study includes both prompt-based and non-prompt-based continual learning methods, ensuring a fair comparison.

Weaknesses:

- **Failure modes and limitations**: The paper does not analyze failure cases (e.g., impact of class imbalance, annotation errors in ODinM-13, or domain shifts).

**Other Comments Or Suggestions:**

No

**Other Strengths And Weaknesses:**

1. Wk and Wv in Eq. 8 are not defined.
2. Lack of theoretical proof for DPA’s efficiency: a formal justification of why DPA outperforms standard prompt attention would improve the rigor of the work.

**Questions For Authors:**

No

**Relation To Broader Scientific Literature:**

This work uses prompt engineering to improve general VLOD models in terms of the medical domain. Its key contribution is related to the decoupling of prompt attention.

**Theoretical Claims:**

No theorectical claim found, which may not be that suitable for ICML.

---

> ### Author Rebuttal · Authors · 2025-04-01
>
> Thank you for the detailed and constructive feedback! We treasure the opportunity to address your concerns and improve our work.
>
> # 1. Mathematical Justification for DPA superiority
> We appreciate the reviewer’s feedback. DPA enhances prompt learning by separating the attention mechanism into distinct prompt-token interactions, minimizing interference from token embeddings. Its key advantage is the re-normalization of attention weights via the scaling factor $\lambda(f_t)$, as derived in Eq. (11-13). This balances the influence of prompts and pretrained tokens, boosting prompt effectiveness when token embeddings dominate due to length. We agree that adding formal theoretical analysis or empirical complexity comparisons would strengthen this and plan to include them in future revisions.
>
> ## Overall Analysis
>
> $$
> f_1 = \text{Concat}[ (1-\lambda(p_t)) \text{Attn}_{v \rightarrow t}(f_v, p_t) + \lambda(p_t) \text{Attn}_{v \rightarrow t}(p_v, p_t);
> (1-\lambda(f_t)) \text{Attn}_{v \rightarrow t}(f_v, f_t) + \lambda(f_t) \text{Attn}_{v \rightarrow t}(p_v, f_t)]
> = \text{Concat}[A, B]
> $$
>
> where $p_{\{v, t\}} \in \mathbb{R}^{l \times d}$ represent the vision and text prompts, and $f_v, f_t$ are the visual and textual features before being fed into $\text{Attn}_{v \rightarrow t}$.
>
> $$
> f_2 = (1- \lambda(f_t)) \text{Attn}_{v \rightarrow t}(f_v ,  f_t) + \lambda(f_t) \text{Attn}_{v \rightarrow t}( p_v,  f_t) = B.
> $$
>
> If $\text{Attn}_{v\to t}(\cdot,\cdot) \in \mathbb{R}^{L \times d}$, then $A(p_t)$ and $A(f_t)$ belong to $\mathbb{R}^{L \times d}$ and $f_1 \in \mathbb{R}^{(L+l) \times d}$, while $f_2 \in \mathbb{R}^{L \times d}$.
>
> $$
> f_2 = \text{Attn}_{v\to t}(f_v,f_t) + \lambda(f_t) \Delta,
> $$
> where $\Delta = \text{Attn}_{v\to t}(p_v,f_t) - \text{Attn}_{v\to t}(f_v,f_t)$.
>
> $$
> \frac{\partial f_2}{\partial \theta} = \frac{\partial \text{Attn}_{v\to t}(f_v,f_t)}{\partial \theta} + \lambda(f_t) \frac{\partial \Delta}{\partial \theta}.
> $$
>
> This indicates that $f_2$ has a lower-dimensional structure, residual components, and a more direct gradient flow.
>
> ## Computation Cost
>
> ### Lemma 1: Computation Cost
>
> **Lemma:**
> $f_2$ is computationally lighter than $f_1$.
>
> **Proof:**
> The overall $f_1$ is $f_1 = \text{Concat}[A;B]$. However, $f_2$ uses only the $B$ branch, $f_2 = B$. The computation cost of branch $A$ is roughly $O_A = O(f_v, p_t) + O(p_v, p_t) =O(A + B)$, while for $B$, $O_{f_2} =O(B)$. Thus, $O_{f_1} >O_{f_2}$.
>
> ## Convergence Benefit Analysis
>
> ### Lemma 2: Convergence Behavior
>
> **Lemma:**
> Let $f_1$ and $f_2$ be models that have converged to local minima, with an optimal representation $f^* = B$. If $f_1$'s output is locally linear around the optimum, then $f_2$ achieves the same performance as $f_1$ at convergence.
>
> **Proof:**
> Consider the loss function $\mathcal{L}(f_{\text{out}})$, where $f_1$ outputs $y_1 = h(\text{Concat}[A;B])$ and $f_2$ outputs $y_2 = h(B)$. At convergence, $\nabla \mathcal{L}(f_1) = \mathbf{0}$ and $\nabla \mathcal{L}(f_2) = \mathbf{0}$, with $f^* = B$ as the optimal representation.
>
> Since $h$ is locally linear at the optima, there exists a matrix $M$ such that:
> $$
> h(\text{Concat}[A;B]) = M \begin{pmatrix} A \\ B \end{pmatrix} = M_1A + M_2B.
> $$
> At convergence, $M_1A = \mathbf{0}$, so:
> $$
> h(\text{Concat}[A;B]) = M_2B = h(B).
> $$
> Thus:
> $$
> y_1 = h(\text{Concat}[A;B]) = h(B) = y_2.
> $$
> Hence, $f_2$ performs as well as $f_1$ at the local minima.
>
> # 2. Lacking evidence in generality beyond ODinM-13
> We thank the reviewer for the suggestion. ODinM-13 already includes tasks across multiple modalities and organs, providing a degree of generalization. To further support generalizability, we conducted additional experiments on polyp datasets from different centers. Results are summarized below:
> | Methods     | Sun   | Kvasir | BKAI  | ClinicDB | FAP ↑ | CAP ↑ | FFP ↓ |
> |-------------|-------|--------|-------|----------|-------|-------|-------|
> | L2P         | 59.22 | 70.01  | 73.16 | 69.24    | 67.91 | 68.69 | 0.35  |
> | DualPrompt  | 62.64 | 71.76  | 75.43 | 72.63    | 70.62 | 69.77 | 1.53  |
> | iDPA (ours)       | 66.10 | 74.33  | 78.77 | 77.93    | 74.28 | 70.92 | -0.03 |
>
> # 3. Failure modes and limitations
> In our experiments, we observed that the IPG module improves learning in low-resource and hard cases. The DPA module further enhances the IPG's performance and helps reduce forgetting. For example, in the CPM-17 dataset (with only 30 training samples), the naive method achieves an FAP of 1.09. Adding only the IPG module improves performance to 15.64, adding only the DPA module achieves 1.80, and combining both modules increases FAP to 36.54.
>
> ## Other notes:
> We will address the missing definitions in Equation (8). Specifically, $W_k$ and $W_v$ represent the linear projection matrices for keys and values in the attention mechanism. These will be clearly defined in the revised manuscript.

---

> > ### Comment · Reviewer_rTU6 · 2025-04-08
> >
> > As noted by Reviewer Sm8X and HWoG, there is still no quantitative comparison with one-for-all style baselines like MedSAM. Plus, I still don't see any FLOPs comparisons with other methods. After considering all the other reviewers' comments, I decided to retain my score.

---

> > > ### Author Response · Authors · 2025-04-08
> > >
> > > We sincerely thank the reviewer for the further feedback. We appreciate the opportunity to clarify the points raised:
> > >
> > > **1. Quantitative comparison with MedSAM**
> > >
> > > We acknowledge the importance of this comparison, as emphasized by the reviewers. We have been trying to reproduce MedSAM2, but due to the unavailability of 2D pre-trained weights and the challenges in adapting 3D weights to 2D data, we were unable to complete the comparison in time. As discussed in our latest conversation with Reviewer Sm8X, we have now added new comparison experiments, which we hope will help address your concerns.
> > >
> > > **2. FLOPs comparison**:
> > >
> > > We appreciate the reviewer’s emphasis on the importance of FLOPs comparisons to assess the computational efficiency of our method. Below, we provide a comparison table for the number of parameters and FLOPs:
> > >
> > > | Methods | #Params↓ | FLOPs↓ |
> > > | --- | --- | --- |
> > > | Joint (Upper) | 231.76M | 488.03 GMac |
> > > | Sequential | 231.76M | 488.03 GMac |
> > > | WiSE-FT | 231.76M | 488.03 GMac |
> > > | ER | 231.76M | 488.03 GMac |
> > > | ZiRa | 10.23M | 490.15 GMac |
> > > | L2P | 6.97M | 601.5 GMac |
> > > | DualPrompt | 4.83M | 583.82 GMac |
> > > | S-Prompt | 2.73M | 590.89 GMac |
> > > | CODA-Prompt | 10.97M | 583.82 GMac |
> > > | DIKI | 8.76M | 583.82 GMac |
> > > | NoRGa | 8.76M | 583.82 GMac |
> > > | iDPA (Ours) | 3.34M | 506.00 GMac/501.00 GMac (train/test) |
> > >
> > > In the case of iDPA, the FLOPs value indicates both training (first number) and testing (second number). As discussed with Reviewer Sm8X, we can merge the key computational parts of the three parallel attentions, which significantly reduces the computational cost of DPA.
> > >
> > >
> > >
> > > We sincerely hope that the additional comparisons and clarifications will provide a clearer view of the contributions and the computational advantages of our method. If it is possible, we kindly ask the reviewer to reconsider the score based on the new information provided.
> > >
> > > Thank you for your continued consideration and valuable feedback. We hope these clarifications address your concerns.

---

### Decision · Program_Chairs · 2025-05-01

**Decision:**

Accept (poster)

**Comment:**

This paper proposes iDPA, a novel framework for incremental medical object detection, introducing two core components: (1) Instance-level Prompt Generation (IPG), which learns fine-grained task-specific prompts from localized visual features, and (2) Decoupled Prompt Attention (DPA), which improves prompt-token interactions and alleviates interference during transfer. The authors validate iDPA across 13 diverse medical datasets in both full-data and few-shot settings and show consistent gains over prompt- and non-prompt-based baselines.

The paper received uniformly weak accept scores from all reviewers. Reviewers generally appreciated the paper's novel framework for incremental medical object detection and its practical value in handling continual learning across diverse medical datasets. Key concerns about lack of comparisons to one-for-all baselines and justifications were raised initially, however, the authors provided a thorough and thoughtful rebuttal, addressing nearly all concerns from reviewers.  The overall consensus leans towards acceptance.